# Fast and Noise-Resilient Magnetic Field Mapping on a Low-Cost UAV Using Gaussian Process Regression

**DOI:** 10.3390/s23083897

**Published:** 2023-04-11

**Authors:** Prince E. Kuevor, Maani Ghaffari, Ella M. Atkins, James W. Cutler

**Affiliations:** 1Robotics Department, University of Michigan, Ann Arbor, MI 48109, USA; 2Naval Architecture and Marine Engineering, University of Michigan, Ann Arbor, MI 48109, USA; 3Aerospace and Ocean Engineering, Virginia Tech, Blacksburg, VA 24061, USA; 4Aerospace Engineering, University of Michigan, Ann Arbor, MI 48109, USA

**Keywords:** Gaussian process regression, magnetic field mapping, unmanned aerial vehicle (UAV), fingerprinting, indoor environments, electromagnetic noise reduction, robust mapping

## Abstract

This study presents a comprehensive approach to mapping local magnetic field anomalies with robustness to magnetic noise from an unmanned aerial vehicle (UAV). The UAV collects magnetic field measurements, which are used to generate a local magnetic field map through Gaussian process regression (GPR). The research identifies two categories of magnetic noise originating from the UAV’s electronics, adversely affecting map precision. First, this paper delineates a zero-mean noise arising from high-frequency motor commands issued by the UAV’s flight controller. To mitigate this noise, the study proposes adjusting a specific gain in the vehicle’s PID controller. Next, our research reveals that the UAV generates a time-varying magnetic bias that fluctuates throughout experimental trials. To address this issue, a novel *compromise mapping* technique is introduced, enabling the map to learn these time-varying biases with data collected from multiple flights. The compromise map circumvents excessive computational demands without sacrificing mapping accuracy by constraining the number of prediction points used for regression. A comparative analysis of the magnetic field maps’ accuracy and the spatial density of observations employed in map construction is then conducted. This examination serves as a guideline for best practices when designing trajectories for local magnetic field mapping. Furthermore, the study presents a novel *consistency metric* intended to determine whether predictions from a GPR magnetic field map should be retained or discarded during state estimation. Empirical evidence from over 120 flight tests substantiates the efficacy of the proposed methodologies. The data are made publicly accessible to facilitate future research endeavors.

## 1. Introduction

The materials used to make buildings create anomalies in the magnetic field that are not modeled by global magnetic field maps (e.g., the World Magnetic Model [1]). Applications that use magnetic fields for indoor navigation must either reject [2] or map [3] these local magnetic anomalies. Mapping the local field requires gathering observations throughout the working volume which can be time consuming if performed by hand [4]. Thus, many researchers instead use robots to spatially sample the local magnetic field. However, the electronics on robots create magnetic noise that reduces map accuracy. Some avoid this electromagnetic noise by separating the magnetometer from other electronics [5,6] or characterizing the vehicle’s noise [7]. In this work, we present new tools to create indoor magnetic field maps using an unmanned aerial vehicle (UAV) while mitigating UAV-induced magnetic noise.

This study, outlined in Figure 1, presents a comprehensive approach to mapping local magnetic field anomalies with robustness to magnetic noise from an unmanned aerial vehicle. The overarching goal of the work is to augment indoor sensor suites with magnetic field data by demonstrating how to create and validate magnetic field maps of indoor spaces. The value of indoor magnetic field maps is demonstrated in works that use them to estimate the position of pedestrians and robotic vehicles inside buildings and to estimate the attitude of a drone indoors. As shown by Ref. [4], gathering observations for a magnetic field map by hand is time consuming and does not scale favorably for large spaces. As such, we propose using an unmanned aerial vehicle to gather observations in the target volume of the workspace, using Gaussian Process Regression (GPR) to interpolate between these observations, then finally leveraging the map in future experiments within the mapped volume.

Unlike humans and wheeled robots, UAVs can easily gather observations of the magnetic field at various altitudes to create three-dimensional magnetic field maps. These maps can then be used to estimate position [8] and attitude [9] indoors. However, UAVs add unwanted magnetic noise, reducing map accuracy and limiting pose estimation accuracy. This paper, outlined in Figure 1, investigates the complexity of using a UAV to create and validate indoor magnetic field maps. Using data from over 100 flights, we analyze how UAV motor commands, distance from electronics, diversity of training points, and spatial density of observations affect the accuracy of magnetic field maps. This work aims to enable fast, accurate, and noise-resilient magnetic field mapping to improve indoor pose estimation techniques.

The contributions of this work are as follows.

Zero-mean noise from the motors and electronic speed controllers (ESCs) can be reduced by removing high-frequency commands from the flight controller.A time-varying magnetic bias caused by the UAV’s electronics is discovered. Although the root cause of this bias is still unknown, techniques are presented to reduce its effect on the resultant map.A novel “compromise map” is introduced that limits the number of observations needed to estimate the magnetic field without sacrificing accuracy.A new consistency test is introduced to assess the credibility of predictions from a magnetic field map. This metric can indicate when one should rely on their magnetometer readings and when another sensing modality should be used instead.

The remainder of this paper is structured as follows. Section 2 presents related work, and Section 3 introduces the mathematical notation for magnetic field maps. Section 4 introduces the UAV and testing methodology for this work. Finally, Section 5 presents results, while conclusions and future work appear in Section 6.

## 2. Related Work

A magnetic field map has an input dimension *p* and output dimension *m*, which we will denote as a p→m map. The choice of input dimension *p* often depends on the agent making the map. For example, p=2 for wheeled robots [10,11] and most pedestrian localization [12] while p=3 for UAVs [13,14], multi-floor pedestrian localization [3], or especially outfitted ground robots [4,15]. The output *m* is set by how the map will be used. Some use the orientation [16] or magnitude [17] of the magnetic field vector with m=1 while others use all three vector components [18,19,20] of the magnetic field with m=3.

Many studies use indoor magnetic field maps to improve pedestrian localization. They often use foot-mounted [21] or calf-mounted sensor suites [22] that can leverage zero-velocity updates. Some papers leverage the ubiquity of smartphones [23] and even consider constructing maps by crowdsourcing magnetic field measurements from many users [24]. Most magnetic field maps for pedestrian localization are 2→m (or “2.5→m” maps [22]), which assume a constant (or near-constant) altitude. Kok et al. present a key exception with a pedestrian-focused magnetic field SLAM (simultaneous localization and mapping) solution tested in a stairwell [3].

Ground vehicles are another common platform for indoor magnetic field mapping [20,25]. These vehicles benefit from wheel encoder odometry [18], but cannot typically create p=3 maps since they are confined to the ground. However, some researchers outfit their wheeled robots with a vertical actuator [4] or vertically spaced magnetometers [15] to make 3→m maps. The latter is a recent work by Hanley et al. that shows indoor magnetic fields are sensitive to altitude changes and emphasizes issues with 2→m planar magnetic field maps [15]. In this paper, we use a UAV to quickly gather observations in p=3 spatial dimensions to make 3 →m magnetic field maps. However, the UAV adds electromagnetic noise that must be mitigated.

UAVs are more frequently used for *outdoor* magnetic field surveys than for *indoor* mapping. A recent review by Zhang et al. [26] compiled papers on outdoor magnetic field surveillance and suppression of UAV-induced magnetic noise. For example, ref. [27] mounted a Geometrics G823A cesium magnetometer on a gas-powered helicopter and measured 800 nT of magnetic variation caused by components on the vehicle. This reduced to 80 nT and 40 nT when the magnetometer was attached to a boom of 0.5 m and 1.2 m, respectively. Similarly, Koyama et al. suspend their magnetometer at the end of a 4.5 m-long cable in [28]. The researchers in [27] considered flight vehicles that cost $2 K–$45 K USD and weigh 7 lbs–51 lbs. Thus, the vehicle size and project budget of outdoor magnetic surveys enable noise-mitigation techniques that are not applicable to the smaller, cheaper UAVs used for indoor mapping.

Some papers use UAVs to leverage indoor magnetic fields without making maps. Brzozowski et al. described methods to support the use of indoor magnetic fields on UAVs but only focus on sampling and visualization techniques [13,14]. The authors in [29] performed 3D position localization of a UAV near power lines but did not attempt to map the magnetic field for their application. Li et al. estimated the 6DOF pose of a UAV indoors using magnetometers (among other sensors), but only used the magnetometer for heading estimates and achieved this without a magnetic field map [30]. Zahran et al. leveraged hall effect sensors in a clever way to estimate the velocity of a quadrotor and improve the dead reckoning of such flight vehicles (again, without magnetic field maps) [31].

Alternatively, some use UAVs to map signals, such as ultra-wideband (UWB) radio [32] and visual light communication (VLC) [33]. However, such papers need not address vehicle-based interference, since UAVs do not add noise to UWB or visual light measurements.

This paper focuses on accurate indoor magnetic field mapping in the presence of UAV-induced noise. UAVs can gather magnetic field observations faster than by hand [4]. Additionally, they can easily obtain observations at various altitudes while ground vehicles require special modifications to do the same [4,15]. The tradeoff is that UAVs add unwanted noise to magnetic field measurements. Similar papers on UAV noise mitigation for indoor magnetic fields use a 40 cm boom to reduce noise and then carefully characterize any remaining noise sources [6,7]. Here, we analyze measured magnetic variation against separation distances of 2cm–8cm and show that adjustments to motor commands can reduce UAV-induced magnetic noise.

The map then incorporates any remaining noise using observations from several flight tests. However, increasing the number of observations used to predict the ambient field increases computation time. Solin et al. leverage Maxwell’s equations for the Gaussian process priors of their 2→3 map [5]. Their model is more accurate and computationally efficient than three independent Gaussian processes and gives real-time predictions using thousands of observations. In this paper, we use three independent Gaussian processes to simplify the modeling and focus on UAV noise mitigation. To address the computational cost of having a larger observation set, we propose a *compromise mapping* technique which effectively limits the number of points used to predict the ambient magnetic field. The proposed compromise map method proved effective for our workspace, but more clever modeling techniques (like in [5]) may be needed to accurately map larger volumes. Additionally, we analyze how the spatial density of observations affects the compromise map’s accuracy with a quantitative version of the analysis performed in [4].

## 3. Mathematical Preliminaries and Methodology

This section introduces Gaussian Process Regression and its use in interpolating a set of magnetic field observations in a workspace. Special notation is used to distinguish a set of $n_{2}$ observations used to train hyperparameters and a separate set of $n_{1}$ observations used to perform inference. Additionally, we introduce performance metrics used later in the analysis of the magnetic field maps.

We first introduce some notation. A single measurement of the magnetic field at an unspecified location is $\tilde{y}\in R^{3}$ with the *x*, *y*, and *z* components of this measurement denoted as ${\tilde{y}}_{x}$, ${\tilde{y}}_{y}$, ${\tilde{y}}_{z}\in R$, respectively. In general, a subscript of ${}_{x}$, ${}_{y}$, or ${}_{z}$ denotes a respective component of the magnetic field while an overhead tilde $\;\tilde{}\;$ denotes a *measured* value. ${\tilde{Y}}^{n}\in R^{3\times n}$ is a set of *n* magnetic field measurements while ${\tilde{Y}}_{z}^{n}\in R^{n}$ denotes the *z* component of each magnetic field measurement in the set. Similarly, $X^{n}\in R^{3\times n}$ is a collection of *n* spatial locations in the workspace. The *predicted* or *estimated* magnetic field $\hat{m}$ at some location $r\in R^{3}$ is denoted as $\hat{m}\left(r\right)\in R^{3}$ and the *x* component of this prediction is ${\hat{m}}_{x}\left(r\right)\in R$. Thus, a collection of *n* magnetic field estimates is ${\hat{M}}^{n}\left(X^{n}\right)\in R^{3\times n}$. Similarly, ${\hat{M}}^{n}$ is a set of predictions of the magnetic field where the location of these predictions is arbitrary and ${\hat{M}}_{y}^{n}\in R^{n}$ gives the predictions from ${\mathrm{GP}}_{y}$. Generally, an overhead hat ^ denotes a prediction/estimate, while a superscript integer *n* denotes the number of measurements/predictions/locations of the respective matrix.

### 3.1. Gaussian Process Regression

GPR is an interpolation tool used to estimate a signal given a set of noisy measurements. Here, we use GPR as the backbone of the magnetic field map leveraging methods from Rasmussen et al. [34]. The goal is to have a magnetic field map of the flight workspace to provide an estimated magnetic field vector at any location in the working volume.

Because GPR is a single output regression method, this work uses GPR to create three separate 3→1 maps, each responsible for the *x*, *y*, and *z* components of the magnetic field. To create the GPR-based map, we first gather *n* observations of the magnetic field throughout the workspace. Observation sets give the three components of the measured magnetic field ${\tilde{Y}}_{x}^{n}$, ${\tilde{Y}}_{y}^{n}$, ${\tilde{Y}}_{z}^{n}\in R^{n}$ at each 3D position in the design matrix $X^{n}\in R^{3\times n}$. Together, these quantities define the training sets $D_{x}=\left(X,{\tilde{Y}}_{x}\right)$, $D_{y}=\left(X,{\tilde{Y}}_{y}\right)$, and $D_{z}=\left(X,{\tilde{Y}}_{z}\right)$ for the *x*, *y*, and *z* GPRs, respectively.

In [34], Rasmussen and Williams define a Gaussian process as a distribution over functions written as
(1)$$f\left(r\right)∼\mathrm{GP}\left(0,k(r,r^{\prime })\right)$$
where 0 is the zero-mean function, *k* is the covariance function (or kernel) and r, $r^{\prime }\in R^{3}$ are 3D positions. This work uses the squared exponential covariance function
(2)$$k\left(r,r^{\prime }\right)=\sigma_{f}^{2}\exp\left(-\frac{1}{2l^{2}}{\left(r-r^{\prime }\right)}^{⊤}\left(r-r^{\prime }\right)\right)$$
with $\sigma_{f}$ as the signal variance and *l* as the length scale. A final term $\sigma_{n}$ is used with Equation (Equation 2) to model the sensor noise. Together, these three terms define the hyperparameters $\Theta =\{\sigma_{f},l,\sigma_{n}\}$ for this kernel. In essence, the squared exponential kernel is a measure of similarity between two 3D locations r and $r^{\prime }$ (scaled by the hyperparameters). It will serve as a weighting term in the inference of the magnetic field at unobserved locations.

Recall that we use three Gaussian processes ${\mathrm{GP}}_{x}$, ${\mathrm{GP}}_{y}$, and ${\mathrm{GP}}_{z}$ to represent the full magnetic field anywhere in the workspace. As such, we need three sets of hyperparameters ($\Theta_{x}$, $\Theta_{y}$, $\Theta_{z}$), each computed using the Gaussian Processes for Machine Learning gpml-matlab toolbox by minimizing the negative log marginal likelihood of the respective training sets $D_{x}$, $D_{y}$, and $D_{z}$. We define hyperparameters optimized over the observations in training set $D_{x}$ as $\Theta_{x}^{*}\left(D_{x}\right)$. (gpml-matlab was created by Carl Edward Rasmussen and Hannes Nickisch http://www.gaussianprocess.org/gpml/code/matlab/doc/ (accessed on 1 September 2021)).

To estimate the magnetic field at some location $r^{*}$, we need a training set *D* and a set of hyperparameters Θ corresponding to a selected kernel *k*. A squared exponential kernel *k* using optimal hyperparameters $\Theta^{*}\left(D\right)$ is denoted as $k_{\Theta^{*}\left(D\right)}\left(r,r^{*}\right)$.

This is the extent of our notation, but an example is helpful. Say we want to estimate the *z* component of the magnetic field ${\hat{m}}_{z}\left(r^{*}\right)$ at some location $r^{*}$. For this, ${\mathrm{GP}}_{z}$ requires a kernel *k* to compare the target location $r^{*}$ against locations $X^{n_{1}}$ in its training set $D_{z}^{n_{1}}=\left(X^{n_{1}},{\tilde{Y}}_{z}^{n_{1}}\right)$. Additionally, its hyperparameters $\Theta^{*}\left(D_{z}^{n_{2}}\right)$ are optimized over a separate set of observations $D_{z}^{n_{2}}=\left(X^{n_{2}},{\tilde{Y}}_{z}^{n_{2}}\right)$. Mathematically, we express this as
(3)$$\begin{array}{cc} {\hat{m}}_{z}\left(r^{*}\right)\; & =E[f_{z}|X^{n_{1}},{\tilde{Y}}_{z},r^{*}]\\ \; & =K_{\Theta_{z}^{*}\left(D_{z}^{n_{2}}\right)}\left(r^{*},X^{n_{1}}\right){\left[K_{\Theta_{z}^{*}\left(D_{z}^{n_{2}}\right)}\left(X^{n_{1}},X^{n_{1}}\right)+\sigma_{n}^{2}I\right]}^{-1}{\tilde{Y}}_{z} \end{array}$$
where ${\hat{m}}_{z}\left(r^{*}\right)\in R$ and matrix $K(X,r^{*})$ has scalar elements $k\left(X^{\left\{i\right\}},r^{*}\right)$ with each $X^{\left\{i\right\}}\in R^{3}$ a column of $X\in R^{3\times n}$.

The verbosity in Equation (Equation 3) specifies an observation set for optimizing hyperparameters $D_{z}^{n_{2}}=\left(X^{n_{2}},{\tilde{Y}}_{z}^{n_{2}}\right)$ and a separate “inference set” $D_{z}^{n_{1}}=\left(X^{n_{1}},{\tilde{Y}}_{z}^{n_{1}}\right)$ for computing similarity weights when predicting the magnetic field at $r^{*}$. This is important because hyperparameter optimization, typically accomplished offline, scales with $O\left({n_{2}}^{3}\right)$ while inference scales with $O\left({n_{1}}^{2}\right)$. With this formulation, the map encodes information from a large set of observations offline, then uses a subset of these observations for inference. We leverage this idea in Section 4.4 to create and query the magnetic field map.

### 3.2. Magnetometer Calibration

To calibrate the vehicle’s magnetometer, we use the model and iterative least squares solver from [35] along with the two-step calibration procedure from [36]. The magnetometer model from [35] is repeated here
(4)$$\begin{array}{c} {\tilde{B}}_{x}=\theta_{a}B_{x}+\theta_{x_{0}}+\eta_{x} \end{array}$$
(5)$$\begin{array}{c} {\tilde{B}}_{y}=\theta_{b}\left({\tilde{B}}_{y}\cos\left(\theta_{\rho }\right)+B_{x}\sin\left(\theta_{\rho }\right)\right)+\theta_{y_{0}}+\eta_{y} \end{array}$$
(6)$$\begin{array}{c} \begin{array}{cc} {\tilde{B}}_{z} & =\theta_{c}(B_{x}\sin\left(\theta_{\lambda }\right)+B_{y}\sin\left(\theta_{\phi }\right)\cos\left(\theta_{\lambda }\right)\\ & +B_{z}\cos\left(\theta_{\phi }\right)\cos\left(\theta_{\lambda }\right))+\theta_{z_{0}}+\eta_{z} \end{array} \end{array}$$
where (${\tilde{B}}_{x}$, ${\tilde{B}}_{y}$, ${\tilde{B}}_{z}$) are measured magnetic field values, ($B_{x}$, $B_{y}$, $B_{z}$) are the true magnetic field, η is sensor noise, and the parameters to solve for are bias ($\theta_{x_{0}},\theta_{y_{0}},\theta_{z_{0}}$), scale factor ($\theta_{a},\theta_{b},\theta_{c}$), and non-orthogonality terms ($\theta_{\rho },\theta_{\lambda },\theta_{\phi }$).

Ultimately, we aim to find parameters $\theta ={\left[\theta_{a},\theta_{b},\theta_{c},\theta_{x_{0}},\theta_{y_{0}},\theta_{z_{0}},\theta_{\rho },\theta_{\lambda },\theta_{\phi }\right]}^{⊤}$ by minimizing the following cost function via an iterative non-linear least squares solver
(7)$$\begin{array}{cc} \Delta B & =B_{R}^{2}-B^{2}\\ \; & =B_{R}^{2}-\left(B_{x}^{2}+B_{y}^{2}+B_{z}^{2}\right)\\ \; & =B_{R}^{2}-g\left({\tilde{B}}_{x},{\tilde{B}}_{y},{\tilde{B}}_{z},\alpha \right) \end{array}$$
where $B_{R}$ is the reference magnetic field strength (taken from the World Magnetic Model [1]) and g() is obtained by solving Equations (Equation 4)–(Equation 6) for ($B_{x},B_{y},B_{z})$.

The key difference in Springmann’s method [35] is to estimate the nine θ parameters in two separate minimization steps, as Wu et al. demonstrated in [36]. We found that this two-step approach gave more consistent results for another magnetometer (not used in this work) on our UAV. As such, we adopted the two-step calibration technique for the primary magnetometer as well.

We start by setting all scaling terms to 1 and non-orthogonalities to 0, leaving the bias terms ($\theta_{x_{0}},\theta_{y_{0}},\theta_{z_{0}}$) in Equations (Equation 4)–(Equation 6). The first minimization computes an optimal set of bias terms ($\theta_{x_{0}}^{*},\theta_{y_{0}}^{*},\theta_{z_{0}}^{*}$) which are used as initial conditions for the second optimization where $\theta_{0}=$ [1,1,1, $\theta_{x_{0}}^{*},\theta_{y_{0}}^{*},\theta_{z_{0}}^{*},$ ${0,0,0]}^{⊤}$. The nine optimal parameters from this second optimization are used as the calibration terms for the magnetometer.

### 3.3. Performance Metrics

We use the root mean squared error (RMSE) of each GPR’s prediction ${\hat{m}}_{x}$ against the corresponding component of the measured field ${\tilde{y}}_{x}$ across a validation set. With this, the RMSE for ${\mathrm{GP}}_{x}$ over some validation set $D_{x}^{n_{v}}=\left(X^{n_{v}},{\tilde{Y}}_{x}^{n_{v}}\right)$ is defined as
(8)$$RMSE_{x}=\sqrt{\frac{1}{n_{v}}\sum_{i=1}^{n_{v}}{\left({\hat{m}}_{x}^{\left\{i\right\}}-{\tilde{y}}_{x}^{\left\{i\right\}}\right)}^{2}}$$
where ${\hat{M}}_{x}^{n_{v}}$ is the *x* component of the predicted magnetic vector, ${\hat{m}}_{x}^{\left\{i\right\}}$ is the *i*th prediction in ${\hat{M}}_{x}^{n_{v}}$, and ${\tilde{y}}_{x}^{\left\{i\right\}}$ is the corresponding *i*th measurement in ${\tilde{Y}}_{x}^{n_{v}}$.

Although we have three GPRs, it is often convenient to summarize the accuracy of their composite estimate as
(9)$$RMSE_{norm}=\sqrt{{RMSE_{x}}^{2}+{RMSE_{y}}^{2}+{RMSE_{z}}^{2}}.$$

## 4. Experimental Procedure

This section introduces our unmanned aerial vehicle, testing locations, and flight trajectories used to construct and validate the magnetic field maps.

### 4.1. Equipment, Facilities, and Setup

All tests were conducted at the Robot Fly Lab in the University of Michigan Ford Motor Company Robotics Building. This indoor flight arena is equipped with eight OptiTrack motion capture cameras with a working volume of 4 m × 3 m × 2.25 m as shown in Figure 2. A ground station computer connected to the OptiTrack system provides ground-truth position and attitude estimates of the vehicle at 120 Hz. The communication setup for the ground station, pilot transmitter, and flight vehicle is the same as that depicted in Figure 3 of [9], but with one less BeagleBone Blue on the UAV.

The motion capture pose estimate is streamed to the flight vehicle in real time but with a communication latency of about 40 ms. For this, the vehicle’s onboard estimate of roll and pitch is used for control while the remaining states (position and heading) are taken from the 40 ms-delayed motion capture packets.

Time synchronization is important to properly associate each onboard magnetometer measurement with a ground-truth pose when creating and validating the magnetic field maps. During flights, our UAV’s BeagleBone Blue is synchronized with the ground station laptop and ground station BeagleBone Green using a Linux tool called chrony, which was created by the Red Hat Software company (https://chrony.tuxfamily.org/ (accessed on 1 February 2022)). In post-processing, this synchronization allows us to associate the non-delayed motion capture data with the drone’s magnetometer readings. Additionally, we use motion capture pose (position and attitude) estimates to rotate the magnetometer data into the world frame and determine the location of each magnetic field measurement.

This work uses time-invariant magnetometer calibration combining techniques from [35,36] (see Section 3.2). Magnetometer calibration is performed outdoors, just South of the University of Michigan’s outdoor netted flight facility, M-Air. This location is far enough away from any buildings that the magnetic field strength is constant over a few meters and should be accurately reflected by the World Magnetic Model (WMM) [1]. For the data gathered on 1 September 2022 (test series t6), the ambient magnetic field reference term used for calibration is $B_{R}=53.1351\;\mu T$ as taken from a WMM online calculator for M-Air’s location at 42.294431${}^{\circ }$ N, 83.710442${}^{\circ }$ W, and 270 m above sea level.

Our flight vehicle “Q1” (Quadrotor one) is shown in Figure 3 and has an RM3100 magnetometer sampled at 200 Hz and an MPU9250 IMU (gyroscope, accelerometer, and magnetometer) sampled at 200 Hz. This work made no use of the magnetometer on the MPU9250, relying solely on the RM3100 for all magnetic field measurements. The BeagleBone Blue (BBB) microprocessor handles communication with the ground station via Xbee radio, logs all sensor data locally, and commands the four motors to achieve the desired flight trajectory. A Turnigy 4S 2650 mAh 20C LiPo battery provides the main power on Q1, while a Turnigy 2S 300 mAh LiPo battery keeps the BBB powered on when swapping 4S batteries between flights.

Figure 3b shows Q1 with different distances from the magnetometer to the rest of the vehicle’s electronics. $S_{1}$ is defined as the distance from the top of the battery mounting plate to the top of the BBB. This distance is $S_{1}$ = 4.79 cm on Q1 and will not change in any experiments through this paper. $S_{2}$ is an adjustable distance defined from the top of the BBB to the bottom of the magnetometer’s mounting plate that allows us to investigate how the magnetometer’s proximity to other electronics affects the consistency of magnetic field measurements on a quadrotor.

Finally, we placed a stationary RM3100 magnetometer in the corner of the flight lab that gathers data every 10 s (0.1 Hz). This stationary sensor is on the ground at approximately (−3.5 m, +2.5 m) in the ($X_{W}$, $Y_{W}$) frame defined in Figure 2 and was used to confirm that the ambient magnetic field remained constant during experiments (Section 5.2.3).

### 4.2. Flight Profiles

Our flight tests are designed around single-altitude scanning patterns that gather observations at a planar slice of the working volume. Figure 4 shows the desired trajectory (solid blue line) has 4 m *x*-axis strides separated by 0.25 m *y*-axis strides. Since all flights begin at the origin (0 m, 0 m), the trajectory in Figure 4 includes diagonal strides to leave and return to the origin at the start and end of each flight test. Finally, some trajectories gather observations at multiple altitudes in a single flight and include *z*-axis strides between each planar trajectory. For multi-altitude tests, the drone traverses a single diagonal stride from the origin to the corner at the first altitude and another diagonal stride from the corner to the origin at the final altitude.

All linear strides are created from quintic spline trajectories that are hand-tuned to enforce a 1.9 m/s speed limit. This is a compromise between minimizing flight time and the limitations of our flight controller.

The 0.25 m *y*-axis spacing is selected to enable a study on how the spatial density of observations affects the accuracy of the resultant map (Section 5.4). A similar study was performed in [4], where magnetic field observations in a 3 m × 3 m × 2.2 m volume are hand-gathered at 0.2 m increments in all axes. For this work, using 0.2 m *y*-axis and 0.2 m *z*-axis spacing throughout the working volume created excessively long flights. Thus, a 0.25 m minimum separation distance is used instead.

#### Flight Profile Nomenclature

Throughout this paper, we refer to flights with ID tags, such as “tY_XX”, where Y refers to the flight test *series* and XX is the two-digit ID of the flight test in that series. This allows the reader to reference our raw experimental data to facilitate the reproducibility of results. Appendix A has more details.

### 4.3. Pre-Processing Magnetic Observations

When creating hyperparameter ($D^{n_{2}}$) and inference sets ($D^{n_{1}}$), we first pre-process the magnetic field data gathered by the RM3100.

Figure 5 shows data from the UAV’s RM3100 for a segment of time when the drone is not moving, and the motors are not spinning. The black dots are raw magnetometer measurements with many spurious measurements that vary by ±3μT for each axis, respectively. We have seen such spurious measurements from three different RM3100 sensors, even one mounted on a platform with no motors or ESCs (electronic speed controllers). Other works using RM3100s did not report such spurious readings [15,37]. Thus, we believe this could be a result of the 200 Hz sampling rate (600 Hz for the whole device to achieve 200 Hz per axis) or something to do with the RM3100’s firmware. Alternatively, this may stem from poor voltage regulation on the BeagleBone Blue. We have not tried sampling from other microprocessors to see if the BeagleBone is the root cause.

Nonetheless, we apply a median filter with a window size of 5 to all RM3100 data in this work. The resultant observations from the median filter are shown in red in Figure 5. Clearly, there are still some outliers in red, but increasing the window size of the median filter delays the signal enough to cause problems for state estimation. Since we want to construct and validate maps with the same pre-processing steps used when performing state estimation, we fix a window size of 5 and work with the remaining outliers.

In addition to a median filter, we downsample the 200 Hz data gathered during the flight test. Since the UAV does not move much in 1/200th of a second, many observations are spatially redundant. As such, we temporally downsample observations to 2 Hz, 4 Hz, or 10 Hz when creating or validating the maps. Figure 4 shows the spatial distribution of 2 Hz and 10 Hz downsampling during a flight test.

Since motion capture data are used to rotate the magnetometer data from the body frame to the world frame, we perform one final check to ensure a timely ground-truth pose estimate. Any temporally downsampled observation with a motion capture pose older than 50 ms is considered “stale” and replaced with the nearest (in time) observation with a timlier motion capture pose. If the “fresh” replacement is already in the downsampled observation set, the redundant observation is removed, leaving one instance in the final set.

### 4.4. Creating and Querying the “Compromise” Map

As mentioned in Section 3.1, a magnetic field map requires each Gaussian process $\{{\mathrm{GP}}_{x}$, ${\mathrm{GP}}_{y}$, ${\mathrm{GP}}_{z}\}$ to have a set of hyperparameters optimized by an observation set $\{\Theta_{x}^{*}\left(D_{x}^{n_{2}}\right),$ $\Theta_{y}^{*}\left(D_{y}^{n_{2}}\right),$ $\Theta_{z}^{*}\left(D_{z}^{n_{2}}\right)\}$ and an inference set $\left\{D_{x}^{n_{1}},D_{y}^{n_{1}},D_{z}^{n_{1}}\right\}$ to compare the target locations. This section explains how we use flight data to create the hyperparameter observation sets (with $n_{2}$ observations) and then use inference from an intermediate magnetic field map to make “compromise” inference sets (with $n_{1}$ observations) where $n_{1}<n_{2}$. The goal of the compromise map is to leverage the flight-by-flight variation in magnetic field measurements (Section 5.2) by training hyperparameters on $n_{2}$ observations from many flights without incurring the computational cost of having a large inference set size $n_{1}$.

The “hyperparameter observation set” for the *y* component Gaussian process ${\mathrm{GP}}_{y}$ is $D_{y}^{n_{2}}=\left(X^{n_{2}},{\tilde{Y}}_{y}^{n_{2}}\right)$ and similar for $D_{x}^{n_{2}}$ and $D_{z}^{n_{2}}$. All three hyperparameter observation sets share the same observed locations $X^{n_{2}}\in R^{3\times n_{2}}$ taken from downsampled observations from any number of *training* flights.

These observations are used to compute $\left\{\Theta_{x}^{*}\left(D_{x}^{n_{2}}\right),\;\Theta_{y}^{*}\left(D_{y}^{n_{2}}\right),\;\Theta_{z}^{*}\left(D_{z}^{n_{2}}\right)\right\}$ by minimizing the negative log marginal likelihood of the observations over the hyperparameters of each respective Gaussian process. The optimal hyperparameters, along with their corresponding observation sets of size $n_{2}$, will serve as an “intermediate” magnetic field map.

From here, we query the intermediate map at $n_{1}$ user-selected locations $X^{n_{1}}\in R^{3\times n_{1}}$ to generate magnetic field observations for the compromise map. The magnetic field “measurements” that complete the inference set $D_{x}^{n_{1}}=\left(X^{n_{1}},{\hat{M}}_{x}^{n_{1}}\right)$ come from the estimated magnetic field values at each location in $X^{n_{1}}$ where
(10)$$\begin{array}{cc} {\hat{M}}_{x}^{n_{1}} & =E\left[f_{x}|X^{n_{2}},{\tilde{Y}}_{x},X^{n_{1}}\right]\;\in R^{n_{1}}\\ \; & =K_{\Theta_{x}^{*}\left(D_{x}^{n_{2}}\right)}\left(X^{n_{1}},X^{n_{2}}\right){\left[K_{\Theta_{x}^{*}\left(D_{x}^{n_{2}}\right)}\left(X^{n_{2}},X^{n_{2}}\right)+\sigma_{n}^{2}I\right]}^{-1}{\tilde{Y}}_{x} \end{array}$$
and similarly for the magnetic field “measurements” ${\hat{M}}_{y}^{n_{1}}$ and ${\hat{M}}_{z}^{n_{1}}$ for inference sets $D_{y}^{n_{1}}$ and $D_{z}^{n_{1}}$, respectively. Here, the intermediate map estimates the magnetic field at $X^{n_{1}}$ by computing their similarity to locations $X^{n_{2}}$. The notation used in Equation (Equation 10) was introduced in Section 3.1.

At last, we have what we call “compromise inference sets” $\left(D_{x}^{n_{1}},D_{y}^{n_{1}},D_{z}^{n_{1}}\right)$ (or simply “inference sets”) made of magnetic field estimates from the intermediate map at.

Our compromise map avoids overfitting to a single training flight by incorporating observations from several flights. Additionally, it performs inference in $O({n_{1}}^{2}n_{q})$ rather than $O({n_{2}}^{2}n_{q})$ where $n_{q}$ is the number of points queried.

## 5. Results and Discussion

This section presents results on identifying and reducing UAV-induced magnetic field noise. Section 5.1 shows that the autopilot’s PID controller creates measurable magnetic field noise that we reduce with targetted adjustments to the gains. Next, Section 5.2 introduces the unusual magnetic biases our UAV injects into the measurements. Here, we show that distancing the magnetometer from the electronics improves the consistency of measurements. We then show that the compromise map (which uses $n_{1}=511$ observations for inference) yields estimates within 0.013 µT of the intermediate map (which uses $n_{2}=2001$ observations for inference) in Section 5.3.

Section 5.4 introduces our spatial density analysis and shows that observations spaced up to 0.55 m apart will yield an accurate indoor magnetic field map. Section 5.5 shows that it is equivalent to either create a specialized map on the norm of the field or simply take the norm of estimates from a vector-valued magnetic field map. Finally, Section 5.6 presents our consistency metric, which aims to identify when a user can rely on predictions from their GPR-based magnetic field map.

### 5.1. Flight Controller Creating Magnetic Field Noise

Our vehicles use the rc_pilot_a2sys autopilot originally forked from the open-source rc_pilot repository by James Strawson and Librobotcontrol (https://github.com/StrawsonDesign/rc_pilot (accessed on 7 May 2019)).

rc_pilot_a2sys uses a four-stage cascaded PID controller as explained in Section IV.B.1 of [38]. The controller in this work started with the gains used in [38]. We then adjusted the gains to reduce noise from the motors and ESCs.

Figure 6 shows magnetometer data from a flight where the UAV (not Q1, but another vehicle of the same construction) flew a single-altitude scanning pattern (Figure 4) at 1.5 m. The data in Figure 6 is in the world frame.

Figure 6a (t5_00) is the magnetometer data with the gains inherited from ref. [38]. Here, the drone moves along the *x* axis (4m stride) of the flight space from 21 s tp 26 s, and again from about 34 s to 40 s. During these time segments, there is a large variance in the measured magnetic field due to the flight controller’s motor commands.

Figure 6b (t5_01) shows another instance of the same trajectory, but with new controller gains. The plots are temporally aligned to easily compare corresponding flight segments. At a glance, it is clear that the variance in the magnetic field measurements is much lower in Figure 6b than in Figure 6a, but not completely gone. This is most evident in the “Mag Y–World” plot of Figure 6b, where the drone moves along the *x* axis (4 m stride) of the flight space from 21 s to 26 s and again from 34 s to 40 s.

As explained in Section IV.B.1 of ref. [38], we use a four-stage cascaded PID controller with an outer loop that operates on position error and an inner-most loop controlling angular rate error. The math for the inner loop is shown below in Equation (Equation 11)
(11)$$\begin{array}{cc} \tau_{\phi } & =K_{4,\phi }^{P}\;\left({\dot{\phi }}_{d}-\;{\tilde{\omega }}_{x}\right)\\ \; & +K_{4,\phi }^{I}\;\int_{0}^{t}\left({\dot{\phi }}_{d}-\;{\tilde{\omega }}_{x}\right)\;ds\\ \; & +K_{4,\phi }^{D}\;\frac{d}{dt}\left({\dot{\phi }}_{d}-\;{\tilde{\omega }}_{x}\right) \end{array},$$
where $\tau_{\phi }$ is the desired torque for roll (ϕ), $\left\{K_{4,\phi }^{P},K_{4,\phi }^{I},K_{4,\phi }^{D}\right\}$ are the *P*, *I*, and *D* gains for the fourth-stage controller along the roll axis, ${\dot{\phi }}_{d}$ is the desired roll rate, and ${\tilde{\omega }}_{x}$ is the measured roll rate from the gyroscope. There are similar fourth-stage PID loops for pitch ($\dot{\theta }$) and yaw ($\dot{\psi }$) rates, respectively.

The ‘noisy’ gains, inherited from [38], used $K_{4,\phi }^{D}=K_{4,\theta }^{D}=K_{4,\psi }^{D}=0.01$ while the modified ‘quiet’ gains set all three of these values to 0. This change alone is responsible for reducing motor-induced noise in Figure 6.

The derivative term of the attitude rate controllers (e.g., Equation (Equation 11) for roll rate) uses numerical derivatives of gyroscope measurements (Figure 7) as part of their computation. As the quadrotor flies, the propellers induce vibrations that make the gyroscope (and accelerometer) measurements rather noisy. Taking numerical derivatives of these noisy gyroscope measurements causes the motors to be commanded with high-frequency inputs that induce measurable electromagnetic noise. We believe this problem is amplified as the total amount of current pulled from the 4S batteries increases. This would explain why the magnetometer variance is larger during the 4 m, *x*-axis stride (with a max commanded velocity of 0.75 m/s for this set of tests) than during the 0.25 m, *y*-axis stride (max commanded velocity of 0.47 m/s).

Aside from changing controller gains, other possible solutions include commanding less-aggressive maneuvers during flight tests, moving the magnetometer further from the motors and ESCs (Section 5.2), or time-varying magnetometer calibration [35]. In addition, from some preliminary data gathered when trying to understand this problem, we believe that differences in motors and ESCs (even those of the same make and model) may create different magnitudes of measurable magnetic noise.

The remainder of this paper uses only the ‘quiet’ controller gains. Please note that the analysis for this section used $S_{2}$ = 2 cm (Figure 3b).

### 5.2. Identifying and Mitigating Flight-by-Flight Variation

The flight controller generates high-frequency magnetic field noise from the motors, but another magnetic anomaly causes *variation* in the measured magnetic field from flight to flight. Variance refers to the spread of points around a mean. In the last section, we reduced the variance of motor-induced magnetic noise by removing derivative gains from a PID loop in the flight controller. To avoid confusion, we use "variation" to describe differences in the magnetic field between subsequent flights of the same trajectory. These differences are not uniformly spread around an average signal, as we will explain in this section.

Section 5.2.1 explains our testing procedure for investigating the flight-by-flight variations, while examples are given in Section 5.2.2. Next, we investigate two sources of the variations in Section 5.2.3 and Section 5.2.4. Finally, we show how distancing the magnetometer from the other electronics reduces the variation in Section 5.2.5.

#### 5.2.1. Testing Methodology

To measure variation across pairs of flights, Q1 flew several repetitions of the 1.5 m single-altitude scanning trajectory (Figure 4). Initially, we believed that each 4S LiPo battery provided a different amount of “resting current” to achieve the same flight maneuver as another 4S battery. For this, each battery started at full charge, and Q1 flew the same 1.5 m scanning trajectory several times in a row. This lets us test whether the flight-by-flight variations were due to the battery’s voltage during a flight. After a few repetitions with the first 4S battery, a new (fully charged) 4S battery was used to power Q1.

Three Turnigy 4S 2650 mAh 20C batteries (denoted as #02, #04, and #14) were used for this analysis. The charge/discharge and storage history of these batteries is largely unknown, as they have been used for several projects since their purchase in May 2021. The only thing we could rigorously control is fully charging each battery to 16.8 V just before a series of flight tests.

For this analysis, each flight was downsampled to 10 Hz or 2 Hz to reduce spatially redundant observations resulting in ∼900 or ∼180 observations, respectively, for each flight. Figure 4 shows the observation set from one flight with 10 Hz and 2 Hz downsampling. Although 10 Hz clearly shows better coverage of the flight space, the RMSE values for this analysis changed by at most 0.03 µT on the $S_{2}$ = 8 cm dataset when training on 10 Hz vs. 2 Hz downsampling sets. Thus, we sometimes use 10 Hz downsampling to illustrate specific points, but typically use 2 Hz downsampling for faster training and validation of the magnetic field maps.

The goal is to test the consistency of magnetic field measurements across pairs of flights. The ambient magnetic field is not likely to change over the course of a few hours. Thus, any variation between pairs of flights can be attributed to UAV-generated magnetic noise.

#### 5.2.2. Examples of Flight-by-Flight variations

By training the map on a single repetition and validating on another, we get data similar to that in Figure 8, which has two types of plots.

For the first type (e.g., Figure 8a), each magnetometer measurement from the *validation* dataset is shown as a red cross. The blue line is the GPR’s predicted mean at the drone’s position. The blue shading around the solid blue line depicts the two standard deviations (2σ) of the GPR’s uncertainty with its prediction.

Figure 8b depicts a black dot as the error between the red cross and blue line while the GPR’s 2σ uncertainty is the gray shading. Here, we plot the absolute value of the error since the sign of the error does not indicate anything of interest. The percentage in the grayscale plots depicts how often the black dots (GPR error) are within the gray shading (GPR’s 2σ uncertainty).

In Figure 8, the map is trained on 2 Hz-downsampled observations from flight t1_09 and validated against 10 Hz-downsampled observations from three flight tests of the same t1 series. These three validation examples are generally representative of the different flight-by-flight magnetic field anomalies we have seen. Figure 8b shows a relatively large steady-state prediction error throughout, Figure 8c has a steady-state error that changes partway through the flight, while Figure 8d is an ideal case where the mapping and validation flights have agreeable measurements.

We refer to this anomaly as “flight-by-flight variation” because the error between pairs of flights is not distributed around some mean bias value. For example, flight t1_09 had similar magnetic observations as t1_15 (Figure 8d) yet starkly different observations than t1_05 (Figure 8b). Further complicating the matter are cases such as t1_04 (Figure 8c) with a time-varying bias.

There is evidence in Figure 8 that suggests these magnetic anomalies are caused by the ESCs and motors. The initial black dots in Figure 8b,c have low error before the anomalous bias takes effect. These beginning points are observations gathered when the drone rests on the ground. This suggests that before the motors and propellers are spinning, the observations from t1_05 and t1_04 are similar to those from t1_09.

This discussion would benefit from an experiment we could not conduct safely. The idea is to constrain the UAV and compare magnetometer data across two cases. One case with the motors off and another where they are spinning. We attempted this without the propellers and found no appreciable change in the bias of the measured magnetic field. Adding propellers increases the current draw required for the motors to maintain a certain angular velocity and more accurately emulates actual flight conditions. However, given our test equipment, we could not safely constrain a UAV whose propellers are spinning.

#### 5.2.3. Changes in Ambient Magnetic Field

This section presents evidence that the magnetic field variation in Figure 8 is not caused by changes in the ambient magnetic field. To measure changes in the ambient magnetic field, we placed a stationary RM3100 magnetometer under a workbench at approximately (−3.5 m, +2.5 m) as defined by Figure 2. This stationary magnetometer was sampled every 10 s (0.1 Hz) by a BeagleBone Blue from July 2022 through February 2023. The BeagleBone is powered by a portable power bank charged at a wall outlet. The power bank allows for continuous measurements through brief power outages in the workspace.

Figure 9 shows approximately 3 h of data gathered during the t6 series of flight tests on 1 September 2022. Raw data are depicted as black dots while median-filtered data (with a window size of five) is shown as a red line. As in Figure 5, there are several spurious measurements that span ±3 µT. In Figure 9, the outliers are cropped to emphasize the average signal.

Figure 9 shows that the mean signal of each axis drifts by no more than 0.2 µT. For reference, the variance of the raw data (including cropped outliers) is 0.13 µT${}^{2}$, 0.09 µT${}^{2}$, and 0.10 µT${}^{2}$ for each respective axis. By contrast, the variations in Figure 8b exceed 1.8 μT for the *z* component of the field. This paper assumes the ambient field is constant over any given day. In practice, it only needs to remain constant over a single flight test series (several hours), supported by the data in Figure 9. With this, we conclude the flight-by-flight variations depicted in Figure 8 are caused by UAV-induced magnetic noise and not by changes in the ambient magnetic field.

The data from Figure 9 was gathered on 1 September 2022, while the data in Figure 8b was gathered on 17 June 2022. We do not have stationary magnetometer data from 17th June (the t1 flight test series) to directly compare with the variation in Figure 8. However, the stationary magnetometer data from September 1st in Figure 5 is generally representative of the ambient magnetic field measurements at that location. Further analysis of the eight months of stationary magnetometer data is presented in Appendix B.

#### 5.2.4. LiPo Batteries and Magnetic Variation

Initially, the data from the t1_XX flight tests (Section 5.2.2) led us to conclude that the flight-by-flight variations were due to differences in batteries. The motors and electronic speed controllers (ESCs) use the most power on the UAV. When electrical current moves through wires, it induces a magnetic field quantified by the Biot–Savart law. In addition, a battery with lower voltage must provide higher current to achieve the same power output. In addition, if we fix the power required throughout a flight test, a battery with lower voltage should source more current than a battery with higher voltage. Together, these properties imply that the induced magnetic field from the UAV’s motors and ESCs should increase as battery voltage decreases. We designed experiments (Section 5.2.1) to investigate this idea. However, the data did not clearly prove or disprove our hypothesis.

During the t1_XX flight tests, each of the three batteries (#02, #04, and #14) flew five consecutive repetitions of the same trajectory (Figure 4). The voltage on battery #04’s first flight ranged from 16.4V to 15.4V while the last repetition ranged from 15.4 V to 14.6 V. There are similar voltage ranges across the five repetitions for batteries #02 and #14. From voltage data alone, we expected all three batteries to behave similarly.

The results show that battery #14 had over 1 µT of variation between its five repetitions, seeming to indicate our hypothesis had merit. Perhaps the gradually decreasing voltage changed the magnitude of the induced magnetic field. However, batteries #04 and #02 had nearly 0.1 µT of variation between their five respective repetitions despite having a similar change in voltage over the five repetitions.

We then thought that battery #14 was less “healthy” than the others. Perhaps its capacitance, impedance, or another property differed noticeably from batteries #04 and #02. If this were true, battery #14 should induce larger variations than the other batteries. However, future tests with battery #14 give pairwise variations of about 0.1 µT as we see with batteries #04 and #02.

Overall, we could not decisively prove or disprove our hypothesis on the relationship between the flight-by-flight variations and battery voltage. Additionally, we did not find a consistent trend between each individual battery and the resultant magnetic variation. For this, we chose to reduce the impact of magnetic variation by moving the magnetometer further from the sources of magnetic noise.

#### 5.2.5. Varying $S_{2}$: Distance from Magnetometer to Electronics

Many works, such as [27,28], demonstrated that creating distance between a magnetometer and onboard electronics can reduce noise. In this section, we show how the magnitude of the flight-by-flight variations decrease as $S_{2}$ increases. Recall that $S_{2}$ is the distance between the magnetometer and the other electronics on the UAV (Figure 3b).

We flew 60 repetitions of the 1.5m-scanning trajectory from Figure 4 over four test sessions (t1_XX, t2_XX, t3_XX, and t4_XX). Each corresponds to a different value of $S_{2}$ (2 cm, 4 cm, 6 cm, and 8 cm). For each test session, we trained the map on the *first* battery #04 flight and validated the map on all flights of the same test session.

Figure 10 summarizes the data of this 60-flight analysis. The horizontal position of each point is the value of $S_{2}$ as 2 cm, 4 cm, 6 cm, or 8 cm. Any horizontal deviations from these discrete values are only for visual clarity and do not reflect any deviation in the actual value of $S_{2}$. The vertical axis is the vector RMSE of all three GPRs (Equation (Equation 9)). Finally, the colors and symbols identify which battery was used for each repetition with a green + for battery #04, an orange △ for #02, and a purple □ for #14.

At $S_{2}$ = {2, 4, 6, 8} cm, each battery flew *N* = {5, 4, 3, 8} repetitions of the 1.5 m scanning trajectory. For $S_{2}$ = 8 cm, each battery started at full charge, flew four consecutive reps, was re-charged, then flew four more repetitions. We depict the second charge of each battery in the $S_{2}$ = 8 cm dataset with a rotated +, △, and □, respectively.

Figure 10 shows that increasing $S_{2}$ makes the observations of each flight more consistent with one another. Since the GPR-based map (for each $S_{2}$) is trained on a battery #04 flight, the scatter plot shows us the magnitude of the flight-by-flight variations relative to an arbitrary battery #04 flight. As we increase $S_{2}$, the variations tend towards a difference of about 0.3 µT to 0.5 µT.

Please note that for each value $S_{2}$, there is a single green + that is validated by the same observations used to train the map. Even so, there are no RMSE values below 0.3 µT. This may be a limit to how accurate maps can be with our testing platform, which would include inaccuracies due to sensor noise (a function of the 200 Hz sampling rate) and any electromagnetic noise the quadrotor generates.

We take a moment to point out a flawed conclusion we stated in our previous work. In Figure 10b of [9], observations from a rectangular trajectory (the first ∼500 points) are validated on a GPR-based map. In that figure, there is about 2 µT of an error on each of the *x* and *z* GPR validation sets except before takeoff and after landing. In [9], we hypothesized that this error was caused by a pitch misalignment in the magnetometer for that single flight (relative to the pitch angle for all the other flights). However, if this were the case, there likely would not be the brief moments of agreement before takeoff and after landing as shown in Figure 10b of [9]. With what we have learned in this work, we believe that was an example of flight-by-flight variation.

Our work in [9] used a different M330 quadrotor whose parts may have produced less magnetic variation between flights. Alternatively, it could be that the nature of attitude estimation (the focus of [9]) is robust to 2–3 µT variations between flights if such anomalies do not significantly change the angle of the ambient magnetic field (which has a magnitude of 40–70 µT in our workspace). Thus, it is likely that these variations were simply less salient in [9] given the previous application of attitude estimation.

Although increasing $S_{2}$ decreased measured variation, we are still left with ∼0.3 µT of variation between pairs of flights. We believe time-varying magnetometer calibration could further reduce flight-by-flight variation. In [35], they use measurements of the electric current near high-powered devices to estimate and offset vehicle-induced magnetic fields from their measurements. However, our vehicle does not have any electric current sensors, so such a solution is out of reach for our platform. Instead, we use the “compromise map” to reduce the influence of these variations on the resultant magnetic field map.

### 5.3. Accuracy of Compromise Map

In this section, we propose a solution to the flight-by-flight variations that incorporates data from several training flights into a single magnetic field map. Since each flight has a chance of giving a different bias in the measured magnetic field, the GPR-based map will overfit if trained on a single flight test. Thus, we instead, use $n_{2}$ observations from many flights that span the working volume to create an “intermediate map” that optimizes hyperparameters $\left\{\Theta_{x}^{*}\left(D_{x}^{n_{2}}\right),\;\Theta_{y}^{*}\left(D_{y}^{n_{2}}\right),\;\Theta_{z}^{*}\left(D_{z}^{n_{2}}\right)\right\}$ and predicts with the $n_{2}$ observations. The compromise map then uses the intermediate map’s estimates at $n_{1}$ user-selected locations to predict the magnetic field using only $n_{1}$ points. The goal of this section is to see how the accuracy of the compromise map varies as a function of $n_{1}$.

#### 5.3.1. Multi-Altitude Trajectories

This section uses data from eight different flight tests listed in Table 1 where the first four flights are typically used for *training* the map while the remaining four are used for *validation*. The trajectories listed in Table 1 are all multi-altitude tests where each single-altitude slice is the same trajectory from Figure 4. “Lower Four alts.” flies at altitudes Z=[−0.5, −0.75, −1.0, −1.25] m, “Upper Four alts.” at Z=[−1.5, −1.75, −2.0, −2.25] m, “Scan-γ” at Z=[−0.5, −1.375, −2.25] m, while “Scan-ϵ” and “Scan-$\epsilon^{⊥}$” fly at Z=[−0.75, −1.5, −2.0] m. The idea is to gather redundant observations throughout the working volume that allow the map to learn the flight-by-flight variations.

Additionally, Table 1 lists the number of 2 Hz-downsampled observations from each flight. The number of 4 Hz and 10 Hz observations are approximately 2× and 5× the values listed in Table 1.

#### 5.3.2. Intermediate Map Accuracy

To start, we demonstrate the value of adding observations from many flights to a GPR-based map. We call maps that are trained on observations from multiple flights an *intermediate map*.

Table 2 gives the performance of a multi-altitude magnetic field map trained on $n_{2}=408$ (2 Hz downsampling) observations from flight test t6_21 and validated on observations from four different flight tests. The second major column lists the vector RMSE (µT) of all three GPRs along with the RMSE of each *x*, *y*, and *z* GPR, respectively. The third major column quantifies how often (as a percentage) an error data point lies within the 2σ uncertainty of the respective GPR. The performance metrics from the second and third major columns are the same as those listed in the grayscale plots of Figure 8.

The map from Table 2 has overfit to observations from flight t6_21. In Section 5.2, we showed that there are flight-by-flight variations in the measured magnetic field. By training on a single flight, we prevent the map from learning to account for these flight-by-flight variations. Please note that training the map on $n_{2}=2038$ observations (10 Hz downsampling) from t6_21 gives norm RMSE values of 0.629 µT, 1.034 µT, 0.751 µT, and 0.552 µT when validating on t6_04, t6_05, t6_06, and t6_20, respectively. Thus, simply increasing the number of training samples from a single flight test does not improve the map’s performance.

By comparison, Table 3 uses $n_{2}=2001$ observations (2 Hz downsampling) from four different flight tests: t6_00, t6_01, t6_03, and t6_21. Please note, in Table 1, that we are now training and validating on one instance each of all four types of flights conducted for this analysis: lower four altitudes, upper four altitudes, scan-γ, and scan-ϵ. By comparing Table 2 and Table 3, we see that adding observations from a *variety* of flights uniformly reduces RMSE and increases the frequency that error falls within each GPR’s 2σ error (usually caused by an increase in each GPR’s uncertainty).

Additionally, training on $n_{2}=9999$ observations (10 Hz downsampling) from the four training flights gives norm RMSE values of 0.497 µT, 0.686 µT, 0.406 µT, and 0.408 µT when validating on t6_04, t6_05, t6_06, and t6_20, respectively. Again, and surprising in this second case, adding more observations from the same ensemble of training flights does not necessarily improve the map’s performance. However, it is clear that training on multiple flights (Table 3) is better than training on a single flight (Table 2).

Aside from the 2 Hz vs. 10 Hz comparisons performed in this section, we do not aim to directly address ways to reduce the cost of training hyperparameters. Instead, we focus on the inference cost $O\left({n_{2}}^{2}\right)$ with what we call a *compromise map*.

The idea is to query the intermediate map at $n_{1}$ user-selected locations $X^{n_{1}}\in R^{3\times n_{1}}$ in the working volume. The intermediate map’s estimates give us a set of “measurements” ${\hat{M}}^{n_{1}}\in R^{3\times n_{1}}$ which (together with $X^{n_{1}}$) form the “observations” used by the compromise map to perform inference. Please note that the compromise map will use the same hyperparameters from the intermediate map for its inference. Section 4.4 gives a more detailed explanation of the process of creating a compromise map from an intermediate map.

#### 5.3.3. Intermediate Map vs. Compromise Map

We now compare the accuracy of the intermediate map, which uses $n_{2}=2001$ observations (2 Hz downsampling) to perform inference, to that of the “compromise” map, which uses only $n_{1}=511$ inference points. Recall that both use the same sets of hyperparameters optimized over $n_{2}=2001$ observations.

The $n_{1}=511$ user-selected locations for this analysis are chosen as follows. Points are distributed evenly through the [−2, 2] m *x* axis span, [−1.5, 1.5] m *y* axis span, and [−2.25, −0.5] m *z* axis span of the working volume. A compromise map location is selected every 0.5 m, 0.5 m, and 0.25 m for the *x*, *y*, and *z* axes, respectively. In total, this gives 504 locations within the working volume. The remaining seven points are evenly spaced from the ground to an altitude of 0.5 m, so the compromise map has some observations during the takeoff and landing sequence above the origin. The number $n_{1}=511$ is an important constraint for another toolbox we used for position localization. Thus, this spatial discretization (0.5 m × 0.5 m × 0.25 m) became a common test state in our work.

Figure 11 is similar in style to the grayscale plots from Figure 8, but with the subplots as three columns rather than as three rows. This format fits more figures on a single page to more easily compare the intermediate and compromise maps.

Figure 11 shows data for the intermediate map on the left column and the compromise map on the right. Here, we see the compromise map has both quantitatively and qualitatively similar performance to the intermediate map despite using nearly a fourth of the points for inference. By comparing the norm RMSE values (in the titles) across the two columns of Figure 11, we see the two maps are never off by more than 0.013 µT (13 nanoTesla) in norm RMSE. This difference is within the noise of the magnetic field measurements of our platform.

For reference, training a map on 408 observations (2 Hz downsample) from t6_21 and validating this map on the same 408 observations gives RMSE values of (0.196 µT, 0.089µT, 0.108 µT, 0.136 µT) for norm, *x*, *y*, and *z* RMSE values, respectively. This means that the map cannot discern differences of 0.013 µT even when it has overfit for the exact observations it will be validated on.

Thus, we take the differences between the intermediate map ($n_{2}=2001$) and the compromise map ($n_{1}=511$) in Figure 11 to be negligible and assert that their RMSE performance is effectively identical.

Of course, a lot of valuable information can be lost by simply comparing RMSE values. However, visual comparison across the two columns of Figure 11 further emphasizes that both the intermediate and compromise maps yield very similar prediction results. The key difference is in the uncertainty of the two maps.

It is easiest to see this in row three (Figure 11d vs. Figure 11e) where the compromise map (right) has spikier gray shading (2σ uncertainty) than the intermediate map (left). The compromise map’s increased uncertainty is primarily due to the choice of prediction locations $X^{n_{1}}\in R^{3\times n_{1}}$ which have 504 points selected from a (0.5 m × 0.5 m × 0.25 m) spatial discretization of the working volume. Recall that each altitude of our validation trajectories traverses lanes separated by 0.25 m (Figure 4). Thus, anytime the compromise map is queried at a location between the (0.5 m × 0.5 m) *x*–*y* points, it will report a higher uncertainty in its predicted magnetic field estimate.

The overall increased uncertainty of the compromise map also explains why the respective GPR errors more frequently fall within two standard deviations of their respective uncertainties. In other words, the plots on the right column of Figure 11 will almost always have higher red percentages than the comparative GPR on the left column given the selected spatial discretization of (0.5 m × 0.5 m × 0.25 m).

This section showed how to avoid overfitting by training a map on data from many flights. It then introduced the *compromise* map to fix the number of prediction points at $n_{1}=511$. We find that the compromise map maintains comparable performance to the intermediate map yet has faster runtime from the fixed number of prediction points. The next section will further investigate how the spatial density of the points $X^{n_{1}}$ affects the performance of the compromise map.

### 5.4. Compromise Map—Spatial Density Analysis

This section seeks to understand how the accuracy of the compromise map changes as a function of the spatial density of the locations used in $X_{1}$. Generally, we expect GPR error to increase, the number of error points captured by 2σ uncertainty to increase, and the computation time to decrease as the training set points become more sparse. This section analyzes the first two trends.

In the last section, the $n_{1}=511$ compromise training points came from a custom spatial density of (0.5 m × 0.5 m × 0.25 m) which gave 504 locations, plus an additional seven locations to have observations along the takeoff and landing segment for each flight. In this section, we work with a uniform spatial density in all directions (*S* × *S* × *S*). To simplify matters a bit, we use a naive linear spacing of [min:S:max] for the locations along each respective spatial axis of the lab. This means that different spatial densities *S* can yield the same number of points $n_{1}$ but at different locations. Recall that our working volume has spatial limits of [−2, 2] m in *x*, [−1.5, 1.5] m in *y*, and [−2.25, −0.5] m in *z*.

We vary *S* from 0.2 m to 1 m to emulate a study performed in [4]. However, our study quantifies the RMSE of the magnetic field map rather than relying exclusively on visual comparisons of the map. For the 17 values of *S* tested (in increasing order), $n_{1}$ = [3031, 1775, 931, 655, 447, 259, 259, 199, 133, 112, 97, 97, 79, 67, 47, 47, 47]. Recall, seven of the $n_{1}$ values listed here are for the takeoff and landing sequence.

Figure 12 shows norm RMSE of the compromise map (validated on t6_04, t6_05, t6_06, t6_20) as a function of the spatial density term *S*. A zoomed-in version of the initial data segment is embedded in the same figure. Here we see that the norm RMSE is nearly constant for values S≤0.5 m with a small spike in norm RMSE at S=0.45 m. This spike is caused by the naive linear spacing, which occasionally causes poor sampling along altitudes. At S=0.45 m, we get altitudes of z= [−2.25, −1.8, −1.35, −0.9] m. Recall from Section 5.3.1 that some validation flights have measurements as low as z=−0.5m.

Combining these sources of information, we can see that the black-× (t6_06; Scan-γ) and red-□ (t6_04; Lower four alts) suffer the worst increases at S=0.45 m. Since the compromise map’s observations only go as low as z=−0.9 m, both these trajectories have higher GPR RMSE when validated at their z=−0.5 m observations. Meanwhile, blue-∘ (t6_20; Scan-ϵ) shows little change at S=0.45 m since all its observations are away from the extremes of the working volume. The other transient spikes in Figure 12 (e.g., S=0.6 m) are caused by similar *z*-axis sampling from the naive linear spacing. t6_04 and t6_06 continue to be the most sensitive to certain values *S* when the map has no observations near z=−0.5 m.

Despite these transient spikes, there are two clear trends we can see in Figure 12. First, for values of S≤0.55 m (with S=0.45 m as an exception), the norm RMSE is relatively insensitive to changes in *S*. This agrees with ref. [4] where they show (visually) that their magnetic field map is qualitatively similar for values of S= [0.2, 0.4, 0.6] m. In [4], their analysis led them to use 0.6 m as a standard distance between observations in a subsequent experiment.

For comparison, we briefly summarize the spatial density of observations used by other indoor magnetic field mapping papers. Haverinen et al. map along a single axis (p=1) and separate observations by 0.04m and 0.1m on a ground robot and chest-mounted pedestrian setup, respectively, [17]. Additionally, [16] makes a p=1 map with 0.2 m between each observation, and [20] make a p=1 map with 0.1 m between observations.

For p=2, [19]] create a constant-altitude planar map using S=0.305 m separation in *x* and *y* for their coarse grid and S=0.005 m for fine grid spacing. This mapping is performed in a 2.1 m × 2.1 m room. Ref. [10] uses an occupancy grid for their p=2 magnetic field SLAM solution with grid size 0.05 m × 0.05 m. The solution ignores all previous magnetic field measurements further than 0.5m from the vehicle’s current position. Wu et al. made a p=2 map with lanes separated by 0.38 m [11]. Hanley et al. created p=2 maps at different heights along hallways in three separate buildings [15]. The authors used a longitudinal separation distance of about 0.4 m (varies per building) and a vertical separation of 0.08 m between their observations.

Finally, Ref. [13] made a p=3 “map” of a hallway (no explicit interpolation of observations) with (0.33 m × 0.45 m ×$S^{?}$) where the vertical spacing $S^{?}$ is not explicitly listed in their work, and [6] made p=3 maps with S=0.5 m uniform spacing in all axes. Our previous work used planar lawnmower patterns, as in Figure 4, but with 0.5 m spacing in the *y* axis (instead of the 0.25 m spacing used in this work) [9]. Furthermore, we used a vertical spacing of 0.75 m in the training sets to investigate the accuracy of p=3 magnetic field maps when interpolating between and extrapolating outside pre-mapped altitudes.

In short, for 1D, 2D, or 3D indoor magnetic field maps, we have not seen any papers using separation distances *S* larger than 0.6 m. The exceptions are [4], where the authors explicitly study a reasonable upper limit on *S*, and our work [9], which studied the interpolative and extrapolative limits of p=3 maps.

We believe the empirical upper bound of S=0.6 m may be driven by proximity to walls and the floor. If this is the case, S≤0.6 m might only be the upper bound for indoor regions that are within a couple of meters from some wall or the floor. We acknowledge that this includes nearly every room or hallway in most buildings, but a large, multi-story atrium might be an exception. Portions of such an open, yet indoor, volume may have lower spatial variation and can be accurately mapped with values S>0.6 m.

It is also possible that the floor, and not walls, are the dominating factor for spatial variation in indoor magnetic fields. In [15], Hanley et al. showed that the magnetic field near the floor (within 0.5 m from the ground) of three university buildings is most distinct from the field at other altitudes up to 2 m. This might explain why the other works in the spatial density summary, which mostly use ground robots and pedestrians that remain “near” the floor, have S≤0.6 m.

The second conclusion, the complement of the first, is that values of S≥0.6 m start to show steady increases in norm RMSE. It is not clear what trends hold for S>1 m, but such separation distances are somewhat degenerate given the size of our working volume. Furthermore, it is unlikely for the norm RMSE values to return to their S=0.2 m for values of S>1 m.

This analysis led us to use S=0.5 m for the *x*–*y* compromise map separation distance. We chose S=0.25 m for *z* because we found (empirically) that the GPR’s performance is most sensitive to omissions in observed altitudes. Recent works on indoor magnetic field mapping have shown similar trends: that indoor magnetic fields change noticeably as a function of altitude [9,15]. For our analysis, similarly for [9,15], it is not clear how much this altitude-based sensitivity is driven by the fact that all flights are comprised of many single-altitude slices.

This section showed that magnetic field map accuracy is similar if observations are within S=0.55 m in all directions. We have not seen any other papers on indoor magnetic field mapping using separation distances larger than S=0.6 m. This upper bound helps to choose locations $X^{n_{1}}$ for a compromise map and informs the design of trajectories when sampling the magnetic field.

### 5.5. Creating a 3→1 Map from Estimates of a 3→3 Map

This section compares two ways of estimating the norm of the magnetic field in a workspace. Some papers create maps on special components of the ambient magnetic field, such as a 2→2 map on the horizontal and vertical components of the field [39,40], or a 1→1 map on the heading (declination) [16]. Alternatively, the norm or heading of the magnetic field can be computed using the components of a p→3 vector map. Here, we see if a specialized 3→1 map provides better magnetic field norm estimates than a composition of the estimates $\sqrt{{\hat{m}}_{x}^{2}+{\hat{m}}_{y}^{2}+{\hat{m}}_{z}^{2}}$ from a 3→3 map.

Here, we use a fourth Gaussian process ${\mathrm{GP}}_{nrm}$ whose measurements are ${\tilde{y}}_{nrm}=\sqrt{{\tilde{y}}_{x}^{2}+{\tilde{y}}_{y}^{2}+{\tilde{y}}_{z}^{2}}\in R$ and provides estimates ${\hat{m}}_{nrm}$. Otherwise, the procedure for training hyperparameters and creating the compromise map for ${\mathrm{GP}}_{nrm}$ is the same as described in Section 4.4.

We define two error terms to compare the two methods of estimating magnetic field norm. One is for the norm composed from the 3→3 vector map (vec) while the other is for the 3→1 norm map (nrm)
(12)$$\begin{array}{cc} e_{vec\_nrm} & =\sqrt{{\hat{m}}_{x}^{2}+{\hat{m}}_{y}^{2}+{\hat{m}}_{z}^{2}}-{\tilde{y}}_{nrm}\\ e_{nrm\_nrm} & ={\hat{m}}_{nrm}-{\tilde{y}}_{nrm}. \end{array}$$

Table 4 lists the RMSE (Equation (Equation 8)) of terms from Equation (Equation 12) across four validation flights. Here, we see both norm estimation methods are equivalent. The difference between the two RMSE values is always less than 1 nT, which falls within the noise floor of the PNI RM3100 on our platform.

This result is reassuring in that there is no need to train a fourth Gaussian process if a user wants the three vector components and the norm of the magnetic field. It is not immediately clear if this result extends to angle-based compositions of the magnetic field, such as declination [16] or inclination.

### 5.6. Consistency Check

Previous sections showed how to reduce (Section 5.1 and Section 5.2.5), identify (Section 5.2.2) and incorporate (Section 5.3) otherwise unmodeled noise from the UAV. However, there may still be unmodeled errors in a magnetic field map from the movement of furniture or if different platforms with their own unique noise sources are used (e.g., a map trained on UAV data but used for pedestrian localization). This section introduces a method to assess the credibility of predictions from a magnetic field map. The goal is to indicate when predictions from a magnetic field map are reliable and when it may be advantageous to leverage a different sensing modality.

Here, we introduce the concept of “consistency” between a Gaussian process map and a validation flight. Given a p→m magnetic field map, we say that a validation flight is *consistent* with the map if all *m* GPRs, respectively, capture 96% of their error within two standard deviations (2σ) of their uncertainty.

The 96% threshold is inspired by the expected number of points that fall within two standard deviations of a univariate normal (Gaussian) distribution. Additionally, the 96% threshold worked well for our test platform, our choice of GPR kernel (squared exponential), the way we optimize hyperparameters (minimization of log marginal likelihood; Section 3.1), and our target application of position localization. Although these kernel and optimization methods are common, we have not investigated how different kernels or hyperparameters influence our definition of consistency.

To demonstrate, we create a $n_{1}=511$ compromise map of 2 Hz-downsampled observations from the following seven flight tests: t6_0, t6_1, t6_3, t6_4, t6_5, t6_6, t6_16. We validate this compromise map on 10 Hz samples from four flight tests (t6_09, t6_11, t6_15, t6_18) to illustrate how we use the consistency metric. Here, we train and validate on different t6_XX tests, comparing them to the previous sections. We frequently use this particular compromise map for studying position localization accuracy. The selected validation flights serve as examples that aid in understanding the consistency check.

Figure 13 shows the map’s performance on the four selected validation flights. We start with a case that easily passes the consistency test (Figure 13a). Test t6_11 fails the consistency test since the ${\mathrm{GP}}_{z}$ does not capture enough of its error within its 2σ uncertainty. Please note that even though ${\mathrm{GP}}_{x}$ and ${\mathrm{GP}}_{y}$ have an accurate understanding of their error, the poor performance of the ${\mathrm{GP}}_{z}$ alone makes test t6_11 inconsistent with this 3→3 map. If the ${\mathrm{GP}}_{z}$ estimates are not needed (e.g., localization using magnetic field heading [16]), then the inaccuracies of ${\mathrm{GP}}_{z}$ may be irrelevant, and the map could still be useful.

Figure 13c shows an example of the flight-by-flight variation changing partway through a validation flight. In this case, the first 1220 validation points easily pass the consistency metric for all three GPRs. From 1220 to 1335, things worsen, and ${\mathrm{GP}}_{x}$ barely fails the consistency test. After data point 1335, all three GPRs fail the consistency check.

Since most of test t6_15 (up to point 1220) is consistent with the compromise map, computing the 2σ percentage across the entire validation set gives (97%, 95%, 95%) for each GPR, respectively. Working with these numbers alone (without having Figure 13c for context), it is tempting to say all of t6_15 is consistent with the map (although barely). However, since flight-by-flight variations can change bias values partway through a flight test, it may be necessary to apply the consistency check to portions of a flight.

Finally, validating on t6_18 gives Figure 13d where the first 1242 validation points undoubtedly pass the consistency test but ${\mathrm{GP}}_{z}$ fails on the remaining points. Please note that in Figure 13d, ${\mathrm{GP}}_{y}$ also changes its behavior near index 1380, but we focus on ${\mathrm{GP}}_{z}$ since it fails first.

Ninety-six percent should be taken as a “rule of thumb” rather than a magic threshold where something fundamentally changes. It is important to consider how the map will be used when deciding whether a validation test should be rejected based on the consistency check alone. Here, for t6_18 in Figure 13d, we also include the RMSE values (in blue) for all points before and after 1242 to illustrate the accuracy of the compromise map before and after the mid-flight bias shift.

For the analysis in ref. [9], though we did not realize it at the time, these flight-by-flight variations were still present. Yet, we used magnetic field maps to improve attitude estimates on a UAV. This may be due to the nature of attitude estimation, which compares the angle of a measured vector against its corresponding reference vector. In [9], the flight-by-flight variations did not significantly change the angle of the magnetic field reference vectors, so they were less of an issue.

Our preliminary results on magnetic field position localization show that the errors in t6_15 (Figure 13c) and t6_18 (Figure 13d) are enough to confuse a particle filter and cause noticeably larger estimation errors. The use of GPR-based magnetic field maps for position localization (via a particle filter) drove the methods throughout this paper and influenced our consistency check threshold. However, if the intended application is more robust to “small” inaccuracies of the map, then a more lenient threshold may work.

## 6. Conclusions and Future Work

This paper presented a comprehensive approach to creating maps of local magnetic field anomalies that are robust to noise from a UAV. Our study demonstrated how UAV motor commands, distance from electronics, diversity of training points, and spatial density of observations affect the accuracy of magnetic field maps. First, we showed that certain portions of a vehicle’s PID controller can increase the measurable magnetic noise when commanding the motors and ESCs on a flight vehicle.

Next, we characterized flight-by-flight magnetic field variations and presented two ways to address them. The first entails moving the magnetometer further from the electronics on the vehicle, while the second aims to inform the GPR-based map about flight-by-flight variations through a “compromise” map. Our findings revealed that the compromise map had similar accuracy to the “intermediate” map, which used four times as many reference points to estimate the magnetic field. Additionally, compromise maps trained on observations spaced S=0.20 m to S=0.55 m apart yield similar accuracy. This finding should inform coverage patterns used to gather observations for magnetic field maps.

Finally, we emphasized that our proposed methods do not encompass all potential mapping errors. Thus, we introduced a “consistency check” as a rule of thumb to assess whether magnetic field observations from a given dataset are consistent with their magnetic field map.

The goal of this work is to make magnetic fields a more accessible sensing modality by creating robust magnetic field maps. We accomplish this using affordable components, relying on real data, and focusing on the accuracy and consistency of the map itself. Our methods and data should reduce the entry cost of learning to use and improving upon magnetic-field-based navigation for indoor systems. In the future, we aim to use these new magnetic mapping techniques for 3D position localization. In line with Section 5.5, we aim to conduct studies similar to [17,39,40] that compare position localization accuracy against the map’s output dimension *m*. We may also revisit the UAV attitude estimation work in [9] with the new mapping methods. Finally, a component-wise causal analysis of the flight-by-flight variations may better help us understand and mitigate the time-varying bias.

## Figures and Tables

**Figure 1 sensors-23-03897-f001:**
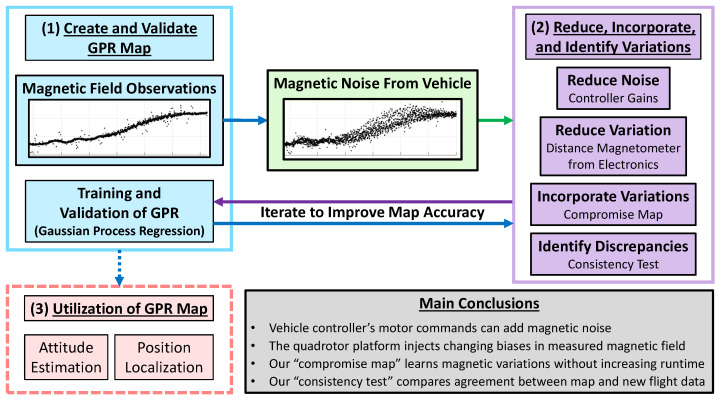
Methods to reduce, incorporate, and identify vehicle-induced magnetic field noise. The proposed interventions can be progressively added until the user’s map is sufficiently accurate for the application.

**Figure 2 sensors-23-03897-f002:**
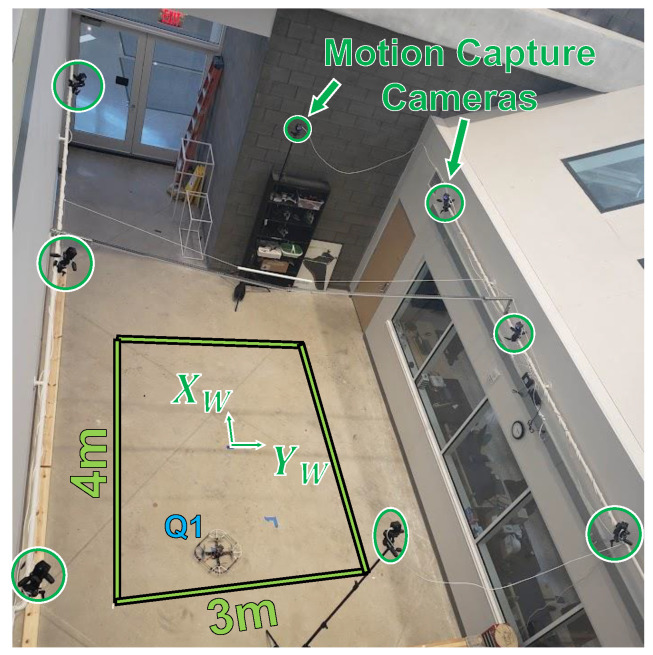
Flight vehicle Q1 in the Robot Fly Lab within the Ford Motor Company Robotics Building at the University of Michigan.

**Figure 3 sensors-23-03897-f003:**
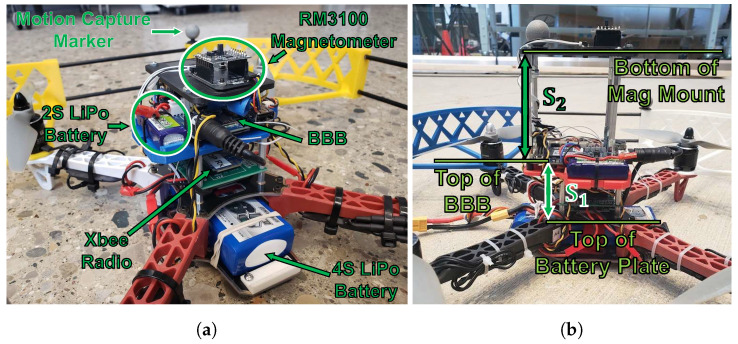
Q1: The flight vehicle used for experiments in this work. (**a**) Main components of Q1. $S_{2}$ = 2 cm. (**b**) Definition of $S_{1}$ and $S_{2}$. $S_{2}$ = 8 cm.

**Figure 4 sensors-23-03897-f004:**
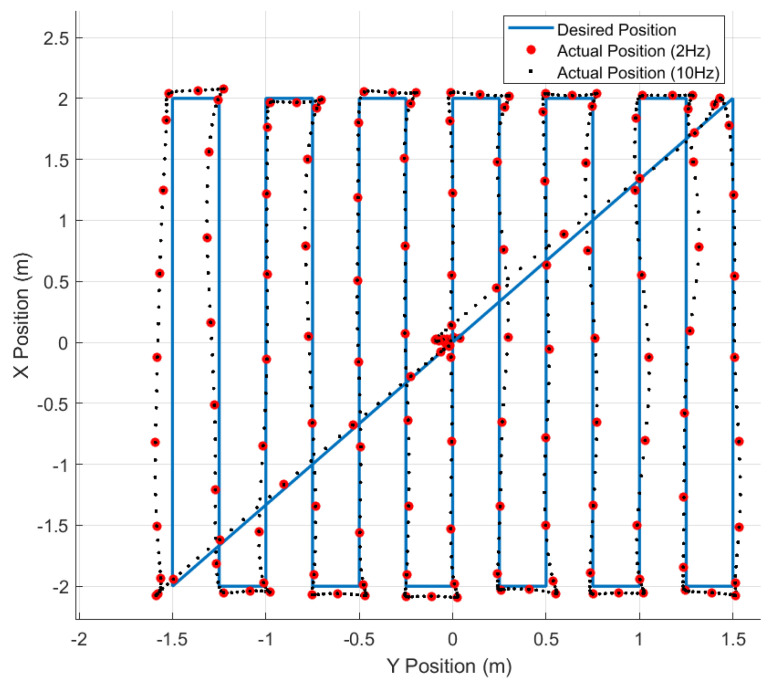
Single-altitude lawnmower trajectory used in all trajectories in this work.

**Figure 5 sensors-23-03897-f005:**
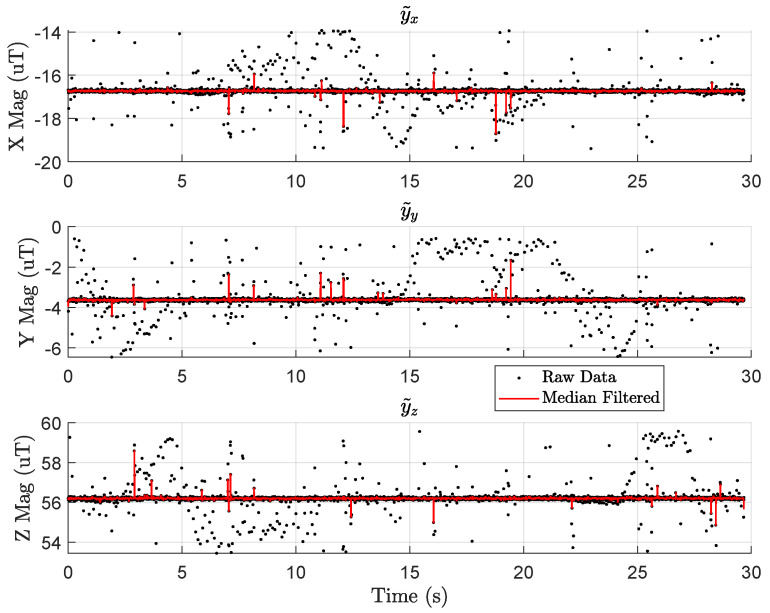
Data from a stationary RM3100 placed at the origin of the flight arena and sampled at 200 Hz. Spurious measurements are reduced with a moving median filter with a window size of 5.

**Figure 6 sensors-23-03897-f006:**
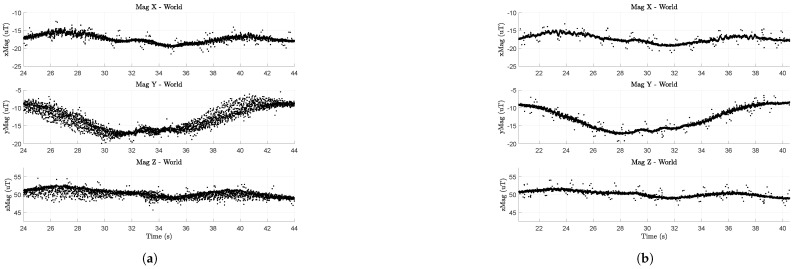
Vehicle “Maggie” flying two instances of the same flight trajectory, showing how the flight controller can inadvertently create noise in the measured magnetic field. (**a**) Test t5_00: Flight with “noisy” gains. (**b**) Test t5_01: Flight with “quiet” gains.

**Figure 7 sensors-23-03897-f007:**
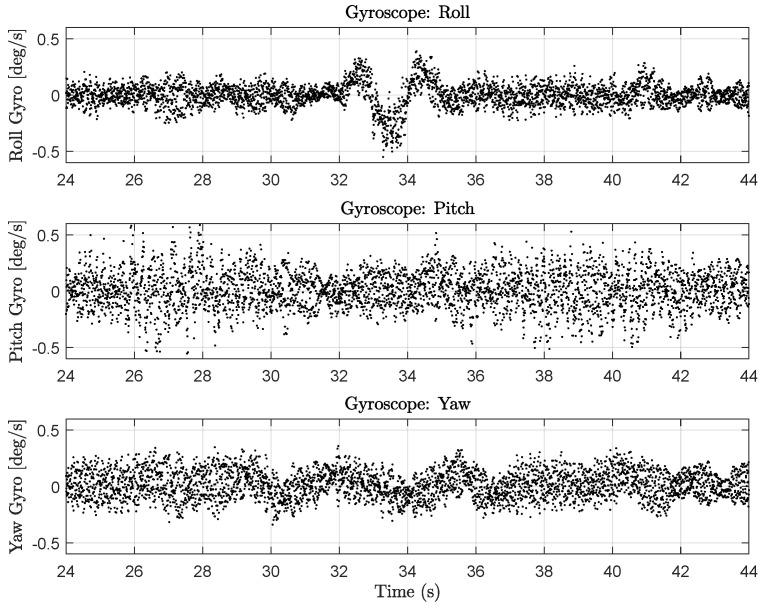
Raw gyroscope data from t5_00. Vibrations from UAV propellers create noisy gyroscope data. Differentiating this noisy data creates the high-frequency magnetic noise seen in Figure 6a.

**Figure 8 sensors-23-03897-f008:**
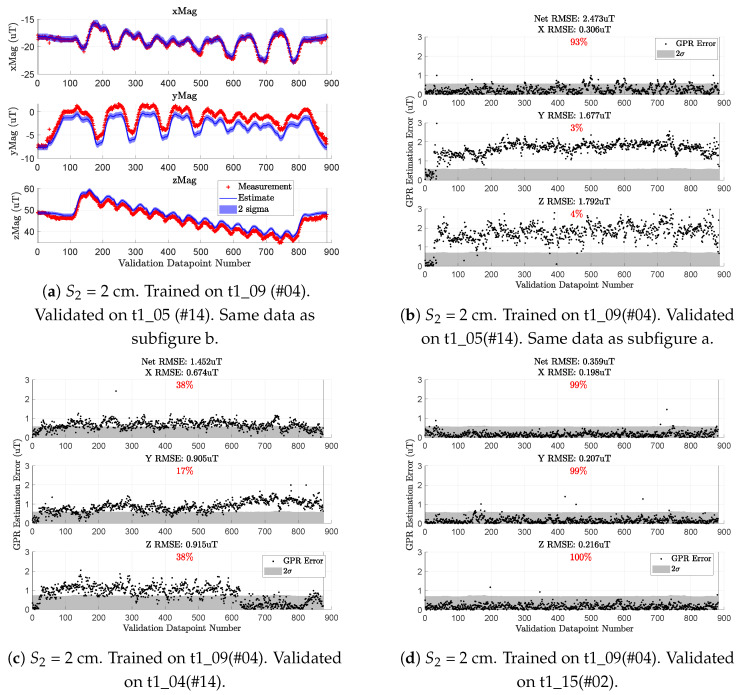
Flight -by-flight variations shown through three validation tests.

**Figure 9 sensors-23-03897-f009:**
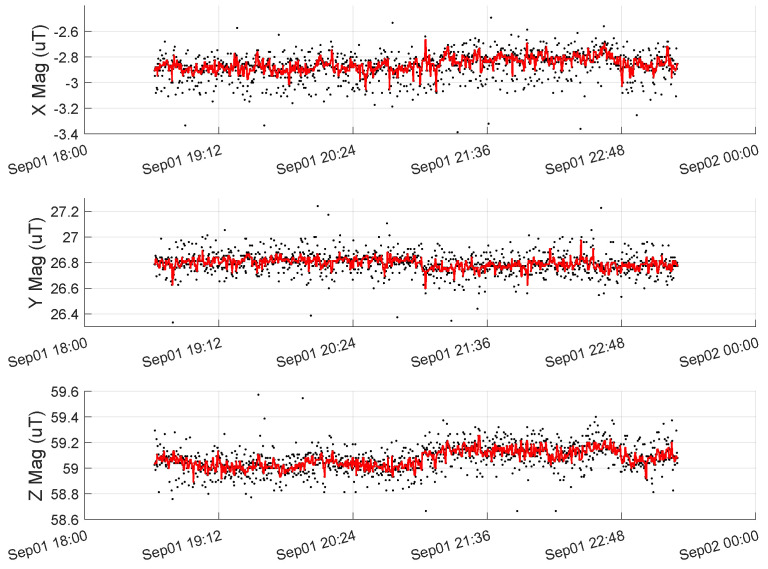
Data from 1 September 2022, during the t6_XX flight series.

**Figure 10 sensors-23-03897-f010:**
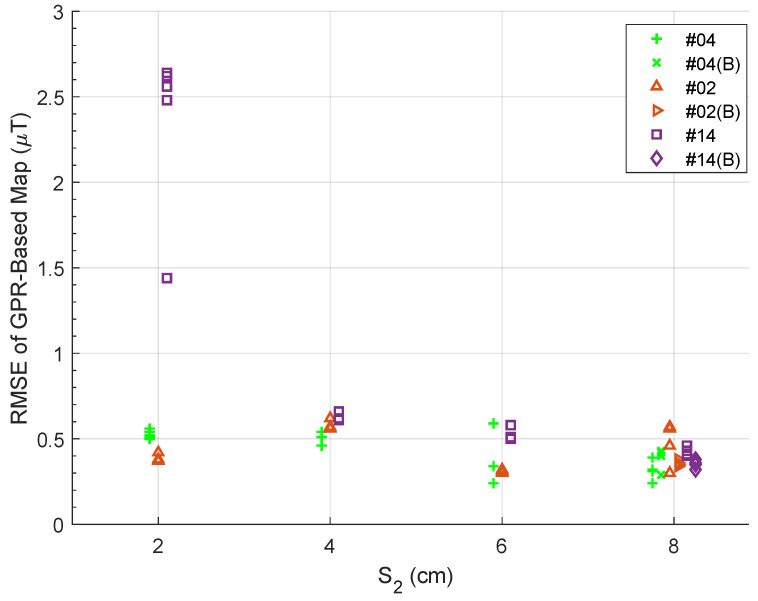
$S_{2}$ versus vector GPR RMSE. At each $S_{2}$, the map is trained on one flight from battery #04 (green +). For $S_{2}$ = {2,4,6,8} cm there are *N* = {5,4,3,8} reps per battery.

**Figure 11 sensors-23-03897-f011:**
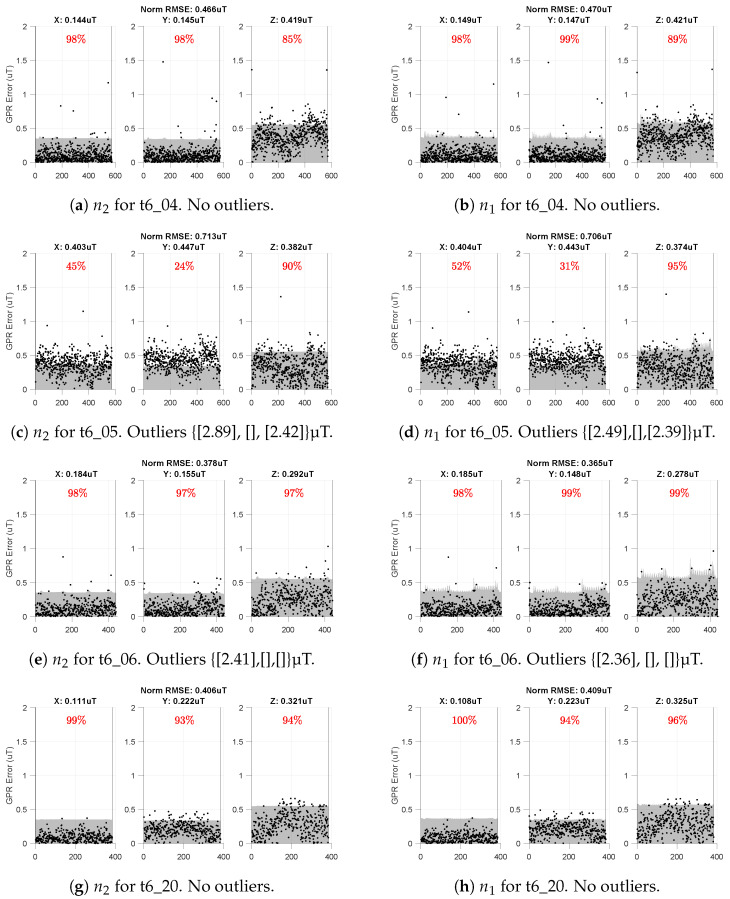
Performance of the intermediate map (**left** column) and compromise map (**right** column). Outliers larger than 2 µT are listed for { [$GPR_{x}$], [$GPR_{y}$], [$GPR_{z}$] } in each subplot.

**Figure 12 sensors-23-03897-f012:**
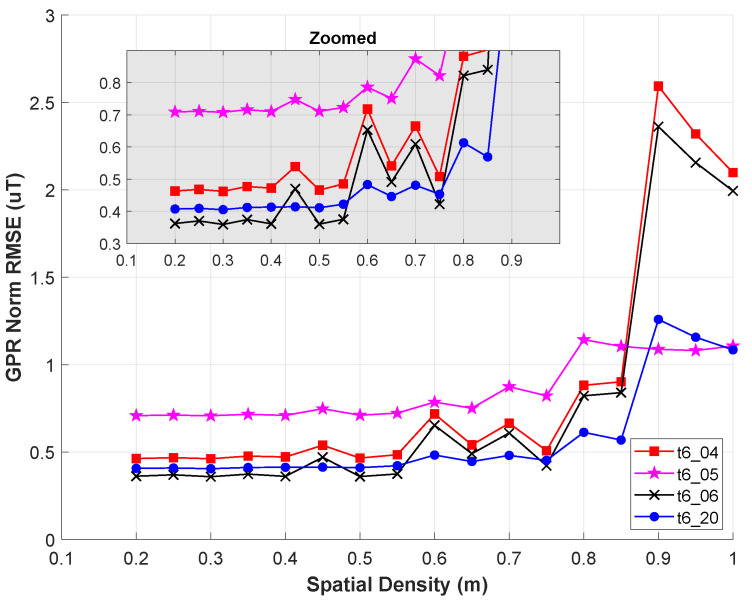
GPR norm RMSE nearly constant for S≤0.55 m. Flights t6_04 (red □) and t6_06 (black ×) have observations at z=−0.5 m causing higher errors for certain values *S*.

**Figure 13 sensors-23-03897-f013:**
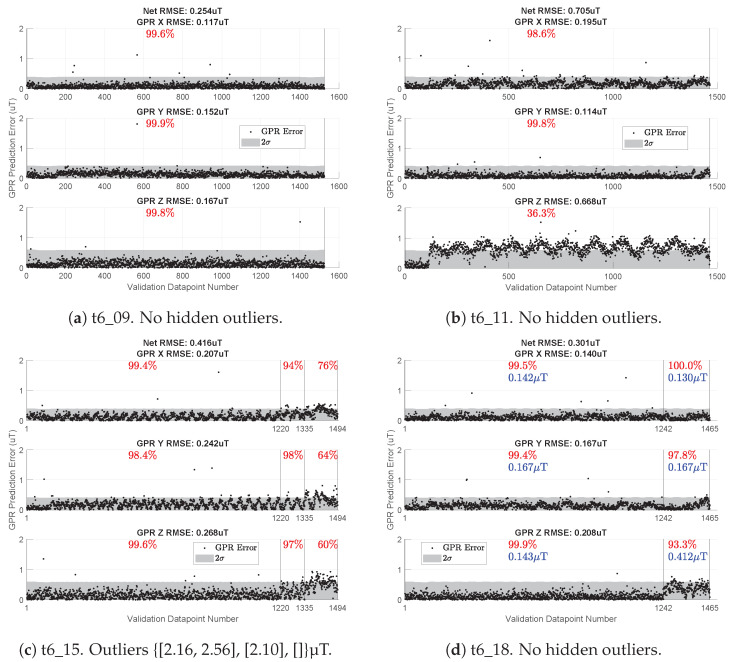
Consistency metric (red percentage) across four validation flights. Some error points are larger than 2 µT and are cut off in the graphs. Such outliers are listed for { [${\mathrm{GP}}_{x}$], [${\mathrm{GP}}_{y}$], [${\mathrm{GP}}_{z}$] } in the respective subplot caption. Empty brackets indicate no hidden outliers for the respective GP.

**Table 1 sensors-23-03897-t001:** Separate flights are used for training and validation. All magnetic field observations were downsampled to 2 Hz for training and validation. The number of observations each training flight contributes to the map is listed.

Flight	Flight	Training [2 Hz]	Validation [2 Hz]
Number	Description	(# Observations)	(# Observations)
t6_00	Lower Four Alts.	571	–
t6_01	Upper Four Alts.	580	–
t6_03	Scan-γ	442	–
t6_21	Scan-$\epsilon^{⊥}$	408	–
t6_04	Lower Four Alts.	–	573
t6_05	Upper Four Alts.	–	573
t6_06	Scan-γ	–	441
t6_20	Scan-ϵ	–	384

**Table 2 sensors-23-03897-t002:** Multi-altitude map error. Trained on $n_{2}=408$ observations from flight test t6_21 (Scan-$\epsilon^{⊥}$).

Flight ID and	GPR RMSE (µT)				Error within 2σ (%)		
Description	*Norm|X|Y|Z*				*X|Y|Z*		
t6_04 (Lower Alts.)	0.595	0.256	0.284	0.456	74.69	87.09	52.88
t6_05 (Upper Alts.)	0.998	0.633	0.541	0.551	6.81	19.72	32.11
t6_06 (Scan-γ)	0.642	0.362	0.318	0.424	56.01	86.62	68.25
t6_20 (Scan-ϵ)	0.556	0.229	0.293	0.413	56.25	47.40	39.06

**Table 3 sensors-23-03897-t003:** GPR error for multi-altitude *intermediate* map trained on $n_{2}=2001$ observations from tests t6_00, t6_01, t6_03, and t6_21.

Flight ID and	GPR RMSE (µT)				Error within 2σ (%)		
Description	*Norm|X|Y|Z*				*X|Y|Z*		
t6_04 (Lower Alts.)	0.466	0.144	0.145	0.419	98.43	98.08	85.17
t6_05 (Upper Alts.)	0.713	0.403	0.447	0.382	45.55	23.73	90.05
t6_06 (Scan-γ)	0.378	0.184	0.155	0.292	97.96	96.83	96.83
t6_20 (Scan-ϵ)	0.406	0.111	0.222	0.321	99.48	92.97	93.75

**Table 4 sensors-23-03897-t004:** Norm (nrm) GPR versus vector (vec) GPR on estimating norm of magnetic field.

Flight ID and Description	*RMSE* (µT)	
	$e_{\mathrm{vec}\_\mathrm{nrm}}$	$e_{\mathrm{hrz}\_\mathrm{nrm}}$
t6_04 (Lower Four Alts.)	0.391	0.390
t6_05 (Upper Four Alts.)	0.256	0.257
t6_06 (Scan-γ)	0.245	0.246
t6_20 (Scan-ϵ)	0.301	0.301

## Data Availability

The data used for this work is available on Zenodo at the following link https://doi.org/10.5281/zenodo.7814818.

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
