# Peer review of "Fast and Noise-Resilient Magnetic Field Mapping on a Low-Cost UAV Using Gaussian Process Regression"

_sensors, 2023, doi:10.3390/s23083897_

Round 1

Reviewer 1 Report

1- An abstract should summarize your project, methods, findings, and conclusions for the reader. The abstract of this work, however, lacks strength and is unintelligible. I recommend revising the abstract to reflect the following ideas: The overarching goal of the article and the research issues you looked at should be brief. The study's fundamental layout. Significant discoveries or trends made as a result of the research. finally, a succinct breakdown of your analyses and findings.

2- The introduction is uninteresting, disjointed, and confusing. It's essential to concentrate on the introduction, convey the events in order, and be readable. Long sentences should not be used since the intended meaning is lost. Consequently, I recommend rewriting the introduction professionally while taking the following to answer the following questions:

Q1: How can you evaluate the presented results according to other studies?’ Prove that the literature review lacks such a study by more modern references.

Q2: ‘What is the importance of the presented paper?’

Q3: What is the main challenge and issues in this study?

Q4: What is the criticism and gap analysis for academic literature that attempts to provide a solution?’

Q5: What are the recommended solutions for such challenges and their issues?’

Q6: What are the present study's implications, contributions, and novelty?’

3- The "Overview of the Proposed Methodology" section is uninteresting, disjointed, and confusing. It's essential to concentrate on the introduction, convey the events in order, and be readable. Rewrite this section and try to list (numbering) the steps to be clearer. Use a flow chart to explain the proposed approach.

4- rewrite the conclusion and consider the following comments:

-   Highlight your analysis and reflect only the important points for the whole paper.

-   Mention the benefits.

-   Mention the implication at the last of this section.

5- Try adding other keywords.

This paper is interesting and valuable, but some minor revisions may be necessary. Please carefully revise my comments.

Author Response

Please read comments in attached PDF

Reviewer 2 Report

Working with real-world data and robotic systems is challenging and time consuming. In this regard, I appreciate the considerable amount of work that the authors put into this study. The contribution of this work can be summarized as:

·      GP regression can be potentially used to model 3D magnetic fields on UAVs.

·      UAVs may cause significant electro-magnetic emissions (EMI). Moving the magnetometer away from the UAV improves the sensors readings.

·      Changing controller gains reduces EMI.

·      The magnetic field readings vary between flights. Training the GP on fields from several flights gives a probabilistic estimate that accounts for these variations.

The current manuscript is in my opinion not ready for publication unless major changes on the manuscripts presentation are done. I value that the authors put a lot of detail in the description of their experiments. Yet, even though many experiments have been carried out, the interpretation of the results lags clarity. In this regard, the presentation is unnecessarily verbose and misses structure. The related work section requires a more detailed discussion of existing research on magnetic field estimation. The description of the GP regression model is imprecise. The discussion of experiments should be further distilled down to the major insights of this study that are also backed up by experimental observation.

Major comments: 

·      C1: The author list the publication [42] “Modeling and Interpolation of the Ambient Magnetic Field by Gaussian Processes“ in their bibliography, but I could not find it being mentioned anywhere in the document. [42] is highly related to the work presented by the authors and must be compared to the work of the authors in the literature discussion. The statement at line 137 “there are only two works that use a UAV […] to make or leverage indoor magnetic field maps” might be technically correct, but could evoke the impression that the authors wish to distract from the fact that there are existing works on mapping indoor magnetic fields with GPs using robots.

·      C2: Equation (2) : The squared exponential covariance function does not include the sensor noise.

·      C3: Notation used for GP regression: In equation (3), you could simply write the equations of a GPs posterior distribution. The chosen notation is not standard and unnecessarily convoluted. Why introduce additional notation “\mathcal{GPR}” instead of explicitly writing down the equations used for making predictions?

·      C4: The fact that you placed an additional stationary magnetometer to check if the field remains constant, indicates that the magnetic field was subject to changes. It would help the reader if you add a sentence on how often the magnetic field changed and also elaborate on the characteristics of these changes. What implications does a change of the magnetic field have on the practical usage of magnetometer-based indoor localization?

·      C5 – Line 547: How is it not safe to collect magnetometer measurements with a quadcopter whose motors are simply switched off?

·      C6– Section 5.2.3: The authors observed that the UAV emits a magnetic field which distorts the magnetometer readings. Then they wonder if this error varies between the different batteries. What I would like to see is an additional investigation of the battery voltage and its effect on the magnetic field error. Afterall, the voltage of a LiPo significantly drops when being discharged which in turn may affect the quadcopter's EMI.

Minor comments:

·      Remark: The computational cost of GP regression growths with O(n^2), sparse GPs keep the computational cost constant but require defining a grid over latent variables which adds additional complexity to the model. Moreover, sparse GPs still require the computing the inverse of a covariance matrix. As alternative approach Gaussian Markov random fields are commonly used for environmental field exploration. The advantage of GMRFs is that you can directly specify the inverse of the covariance matrix while adding additional measurements to your field belief at a constant computational cost. If your environmental field changes over time, you might want to combine your GP/GMRF modeling the spatial field belief with a Kalman filter that accounts for the fields temporal changes.

·      Remark: A magnetic field is typically subject to physical laws that one could use to build structure into the GP kernel subsequently improving the models data-efficiency and prediction accuracy, see e.g. Jidling, Carl, et al. "Linearly constrained Gaussian processes." Advances in neural information processing systems 30 (2017).

·      Remark – Line 183: “Gaussian process regression is a machine learning tool” – From a mathematical point of view, a Gaussian process is a generalization of the Gaussian probability distribution defining a distribution over functions.  GPs find widespread adoption in many different fields including statistical modeling, system identification, and in machine learning. These fields are related and often use similar algorithms.

·      Writing style – Line 137: “…to make or leverage indoor…”, using two verbs in one sentence is unnecessarily verbose.

·      Writing style – Line 513 to 515: I am unsure, if it serves the reader to describe how a Figure literally looks like.

·      Writing style - Line 159:  This sentence seems to not fit into a section about related work. I think it will suffice to briefly mention the fact that position measurements are erroneous when you introduce the GP model.

·      Writing style – Line 388: “GPR maps estimate” you could simply write “GPs”. Your GP models the magnetic field, GP regression is the process of using linear algebra to compute the posterior distribution of your GP. Also, using 3 independent GPs for each output dimension, simply corresponds to using a single multi-dimensional GP with a diagonal as well as matrix-valued covariance function.

·      Writing style – Line 521: “The second type of plot (e.g. Figure 8b)” replace by “Figure 8b depicts…”

·      Writing style – Line 581: “Figure 10 shows two things”, Instead of saying that a figure shows something, you can directly state what you would like the reader to know.

·      Writing style – 583: Avoid the use of “really” in scientific writing.

·      Check your bibliography.

Author Response

Please read comments in attached PDF

Reviewer 3 Report

The topic is interesting. The contribution is clear. The paper is well-written. However, there is some improvements are required to address:  

- What is the research gap and research statement? 

- The author should justify the decision made in work such as the selected methods, performance assessment, and experimental settings. 

- What are the practical implications of the work? it should be stated in the conclusion section. 

Author Response

Please read comments in attached PDF

Reviewer 4 Report

The authors presented the methods to reduce the UAV-INDUCED noise and map the magnetic field with GPR. It is interesting and useful. However, in the manuscript, the detail and process of GPR model should be enhaced before publication, such as the parameters selection, the data size and traing and testing process.

Author Response

Please read comments in attached PDF

Round 2

Reviewer 2 Report

The readability of the paper has significantly improved. The new figures are a nice addition to this study.

- The abstract could still be a bit shortened to efficiently communicate your core ideas to potential readers.

- The grammar of the first sentence in Section 1 is a bit off. You can rewrite the sentence to "Global magnetic field maps (...) do not model anomalies that are caused by commonly used building materials." As a general rule of thumb, if a sentence can be split up into two sentences or shortened without reducing its information content, I would do so.

- The link in line 185 could be a footnote.

- Kindly consider adding conditional color formatting to Table 2.

- In Table 2 and Table 3, you need to specify the units.

Reviewer 3 Report

Authors did all comments.  The manuscript is accepted in the current form.